# Vimentin filaments interact with the actin cortex in mitosis allowing normal cell division

Sofia Duarte [1], Álvaro Viedma-Poyatos [1], Elena Navarro-Carrasco [1], Alma E. Martínez [1], María A. Pajares [1] & Dolores Pérez-Sala [1]

The vimentin network displays remarkable plasticity to support basic cellular functions and reorganizes during cell division. Here, we show that in several cell types vimentin filaments redistribute to the cell cortex during mitosis, forming a robust framework interwoven with cortical actin and affecting its organization. Importantly, the intrinsically disordered tail domain of vimentin is essential for this redistribution, which allows normal mitotic progression. A tailless vimentin mutant forms curly bundles, which remain entangled with dividing chromosomes leading to mitotic catastrophes or asymmetric partitions. Serial deletions of vimentin tail domain gradually impair cortical association and mitosis progression. Disruption of f-actin, but not of microtubules, causes vimentin bundling near the chromosomes. Pathophysiological stimuli, including HIV-protease and lipoxidation, induce similar alterations. Interestingly, full filament formation is dispensable for cortical association, which also occurs in vimentin particles. These results unveil implications of vimentin dynamics in cell division through its interplay with the actin cortex.

---

[1] Department of Structural and Chemical Biology, Centro de Investigaciones Biológicas (CSIC), Ramiro de Maeztu 9, 28040 Madrid, Spain. Correspondence and requests for materials should be addressed to D.P.-S. (email: dperezsala@cib.csic.es)

The vimentin filament network provides architectural support for cells and contributes to the positioning and function of cellular organelles[1–3]. Vimentin plays multiple roles in cell regulation by interacting with signaling proteins, adhesion molecules[4,5], chaperones[6,7], and other cytoskeletal elements[8,9]. The vimentin monomer consists of 466 residues organized in a central rod of predominantly α-helical structure flanked by intrinsically disordered N- and C-terminal domains. Vimentin polymerization is believed to progress from parallel dimers to antiparallel tetramers, eight of which associate laterally

**Fig. 1** Tailless vimentin(1-411) disrupts vimentin wt distribution and interferes with chromosomes in mitosis. **a** Scheme of vimentin domains with the tail sequence displayed in full. **b** Scheme showing the experimental strategies: bicistronic plasmids coding for DsRed2 fluorescent protein (RFP) and untagged vimentin wt (RFP//vim wt) or tailless (residues 1-411) (RFP//vim(1-411)) were transfected into vimentin-expressing cells, vim(+), or vimentin-deficient cells, vim(−), alone, for detection by immunofluorescence, or together with a small amount of the corresponding GFP-vimentin construct (GFP-vim) for direct visualization. **c** SW13/cl.2 human adrenocarcinoma, MCF7 breast carcinoma and HAP1 vim(−) cells transfected with the indicated constructs were observed live 48 h later. The graph shows the proportion of the cellular area occupied by every construct (*$p < 10^{-5}$ vs. wt). **d** SW13/cl.2 cells were transfected with the indicated amounts (in μg) of vim(1-411) (upper panels), or with different proportions of constructs coding for vimentin wt (yellow) or (1-411) (pink), as detailed in the "Methods" section (lower panels), and vimentin condensation was measured as above (*$p < 10^{-5}$ vs. 10:0 vim wt:vim(1-411); #$p < 0.05$ vs. 0.2 and 0.8 μg). **e** U-251 MG astrocytoma, Vero and SW13 parental cells were transfected with RFP//vimentin wt or RFP//vimentin(1-411). Full-length vimentin condensation was assessed by immunofluorescence with V9 anti-vimentin antibody, which recognizes the tail domain (green) (*$p < 10^{-7}$ vs. wt). **f** SW13/cl.2 cells were transfected with RFP//vimentin wt or (1-411) and vimentin distribution assessed by immunofluorescence. Single overlay sections are shown. **g** SW13/cl.2 cells were visualized live after transfection with RFP//vimentin plus GFP-vimentin wt or (1-411), as indicated. CFP-lamin A was used to delimit the nuclear envelope. Insets in **f** and **g** display overall projections of merged images. **h** SW13/cl.2 cells were transfected with RFP//vimentin wt or (1-411) and vimentin distribution in mitosis was observed by immunofluorescence. Single sections taken at mid-cell height (left images) and 3D-reconstructions (right images) are shown. Images in small panels below depict overall projections for vimentin alone (left) or the merge of the three channels (vimentin, RFP and DAPI). Scale bars, 20 μm. The number of determinations for the experimental conditions shown in graphs from left to right was the following: **c** 20, 22, 25, 23, 26, 36; **d** 20, 20, 20, 20, 11, 16, 13; **e** 20, 20, 36, 52, 30, 30. Average values ± SEM are shown. All $p$ values were obtained with two-tailed, unpaired Student's $t$-test and original datasets are provided as Source Data file

in unit length filaments (ULF) that engage head to tail to form filaments. The vimentin network is highly dynamic and rapidly responds to heat-shock, oxidative and electrophilic stresses, ATP and divalent cation availability[10,11], playing a key role in cell adaptation. This fast and versatile remodeling relies on the exchange of subunits or filament segments, as well as on post-translational modifications[12]. Oxidative and electrophilic modifications of vimentin induce drastic alterations in filament architecture and network reorganization, in which the single cysteine residue plays a crucial role[11,13,14]. In addition, phosphorylation of specific residues regulates vimentin assembly and involvement in migration and invasion[15,16], as well as reorganization in mitosis[17,18].

Whereas the role of the vimentin head domain in filament assembly and reorganization has been thoroughly studied[19–21], the role of the tail domain is still incompletely understood. Purified tailless vimentin (vimentin(1-411)) polymerizes in vitro into nearly normal filaments[22–24], presenting oligomerization and sedimentation behaviors similar to those of full-length vimentin[25], although higher heterogeneity and wider average diameter have also been noted[26,27]. In turn, the vimentin tail has been suggested to undergo conformational changes during filament elongation and assembly in vitro[28], and to modulate interactions with divalent cations[22,29].

In cells, vimentin(1-411) mutants form either normal extended arrays or filaments with a tendency to collapse, depending on the experimental system[23,24]. Additionally, the tail domain has been proposed to act as a cytoplasmic retention signal[30] and contribute to filament stability[23]. However, the mechanism(s) by which C-terminally truncated vimentin forms induce cellular perturbations has not been fully elucidated.

Here, we have addressed the importance of the tail domain in vimentin organization and remodeling in several cell types. Our results soon revealed that C-terminal-truncated mutants exert deleterious cellular effects, forming juxtanuclear bundles that snare the mitotic apparatus during cell division. This led us to monitor full-length vimentin showing that vimentin filaments redistribute to the cell periphery in mitosis, in close interplay with the actin cortex, leaving ample space for dividing chromosomes. Moreover, the presence of vimentin affects the organization of cortical actin, thus positioning vimentin as an element of the cell cortex in mitosis, in a tail domain-dependent manner.

## Results

**Vimentin tail is required for correct network distribution**. To elucidate the roles of the vimentin tail domain (Fig. 1a) in filament assembly and stress responses in cells, we employed several strategies, schematized in Fig. 1b, to express vimentin wild type (wt) or vimentin(1-411) in a number of vimentin-positive or vimentin-deficient cells (detailed in Supplementary Table 1). Among vimentin-deficient models, we used: adrenal carcinoma SW13/cl.2 cells, a well-known model for vimentin cellular studies due to their lack of cytoplasmic intermediate filaments[31]; MCF7 breast carcinoma cells, which express various cytoplasmic keratins but no vimentin[32]; and the fibroblast-like HAP1 *VIM* knockout cell line, HAP1 vim(−). These cells allow dissecting the effects of vimentin mutations without interference from the endogenous protein. For direct live network visualization, cells were co-transfected with RFP//vimentin bicistronic plasmids, expressing untagged vimentin, plus a tracer amount of GFP-vimentin vectors (Fig. 1b). Whereas vimentin wt formed an extended network in vimentin-deficient cell lines, the organization of vimentin(1-411) was drastically altered, typically forming curly juxtanuclear filament bundles (Fig. 1c). In SW13/cl.2 cells, vimentin(1-411) bundles were observed even after transfection with low amounts of plasmid (Fig. 1d). Notably, when co-transfected together with vimentin wt, low levels of vimentin(1-411) incorporated into a normal network, whereas, if transfected in excess over wt, it impaired vimentin wt network extension and induced its condensation into coiled bundles (Fig. 1d). Therefore, we explored the impact of overexpressing vimentin(1-411) in several cell types expressing endogenous vimentin (Fig. 1e). For this, we employed: U-251 MG astrocytoma cells, which besides vimentin, express other cytoplasmic intermediate filament proteins including GFAP, nestin, synemin, and keratins in variable proportions[33]; fibroblast-like Vero cells, which also express various intermediate filament proteins; and parental SW13 cells, which express vimentin as the only cytoplasmic intermediate filament[31]. Importantly, overexpression of vimentin(1-411) but not vimentin wt, markedly disrupted the endogenous network in all three cell types, leading to different aberrant patterns, which included filament retraction from the cell periphery and peri-nuclear or juxtanuclear condensation (quantitated in Fig. 1e). Together, these results illustrate that, although vimentin(1-411) polymerization is not impeded, its cellular organization is severely altered. Moreover, vimentin(1-411) exerts deleterious effects on

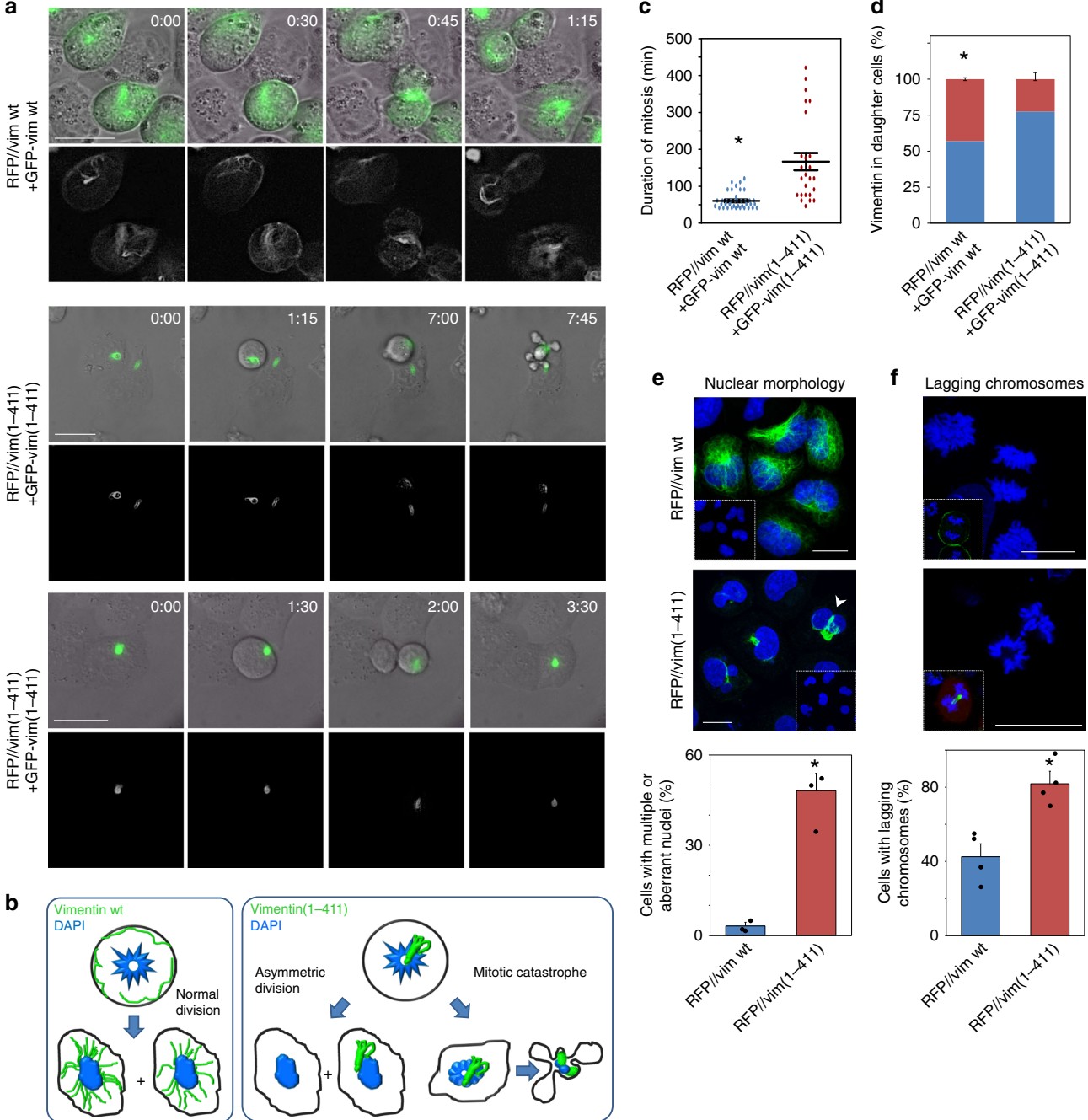

**Fig. 2** Monitorization of cells expressing vimentin wt or vimentin(1-411) during mitosis. **a** SW13/cl.2 cells were transfected with RFP//vimentin wt plus a tracer amount of GFP-vimentin wt, or the equivalent constructs for vimentin(1-411), as indicated, for live cell monitoring by time-lapse microscopy. Several fields were randomly selected and images were acquired every 15 min. Representative images of the overlays of DIC and vimentin green fluorescence (upper panels) and green fluorescence only (lower panels, in gray scale), at the indicated time points, are shown. Scale bars, 20 μm. **b** Schematic representation of the main fates observed for cells transfected with each construct. **c** Duration of mitosis from cell rounding to separation of daughter cells (* $p < 10^{-6}$). **d** The proportion of vimentin signal present in each daughter cell is presented in different colors (* $p < 10^{-5}$). **e** SW13/cl.2 cells were transfected with RFP//vim wt or (1-411). Nuclear morphology was assessed by DAPI staining and vimentin distribution by immunofluorescence. The percentage of cells showing multiple or aberrant nuclei (arrowheads) is depicted in the lower graph (* $p < 0.002$). **f** SW13/cl.2 cells, transfected as in **e**, were synchronized by mild nocodazole treatment. Cells in anaphase were monitored 100 min after nocodazole removal and the proportion of cells showing lagging chromosomes, assessed by DAPI staining, is depicted in the lower graph (* $p < 0.01$). The number of determinations for the experimental conditions shown in graphs from left to right was the following: **c** 38, 25; **d** 24, 16; **e** 3 totaling 336 cells, 3 totaling 219 cells; **f** 4 totaling 96 cells, 4 totaling 64 cells. Average values ± SEM are shown. All $p$ values were obtained with two-tailed, unpaired Student's $t$-test and original datasets are provided as Source Data file

the organization of full-length vimentin, which depend on the proportion of both forms.

**Vimentin peripheral relocation in mitosis is tail-dependent.** Detailed observation of the juxtanuclear structures in SW13/cl.2 cells showed that while vimentin wt filaments extended outwards from the nuclear periphery, vimentin(1-411) thick bundles displayed extensions into the area covered by DAPI staining (Fig. 1f), appearing in deep invaginations of the nuclear envelope or distributing between nuclear lobules (Fig. 1g, arrowheads). This drove us to assess vimentin(1-411) behavior in mitotic cells. In SW13/cl.2 cells, vimentin wt did not disassemble in mitosis but remained as robust filaments with a marked peripheral distribution (Fig. 1h, left panels). In sharp contrast, vimentin(1-411) remained tightly packed in coiled bundles, frequently in close proximity of condensed chromosomes or forming loops encircling them (Fig. 1h, right panels).

**Expression of tailless vimentin leads to aberrant mitosis.** The striking pattern of vimentin(1-411) in the proximity of dividing chromosomes prompted us to monitor its impact on mitotic progression. Time-lapse experiments showed that SW13/cl.2 cells transfected with vimentin wt divided regularly, with an interval between cell rounding and daughter cell separation of 1–2 h (Fig. 2a, upper panels, and Supplementary Movies 1 and 2), yielding nearly homogeneous vimentin distribution between daughter cells. Conversely, cells harboring vimentin(1-411) bundles suffered diverse perturbations. Some cells attempted to divide for several hours and died during or shortly after mitosis, undergoing extensive membrane blebbing, typical of mitotic catastrophe (Fig. 2a, middle panels, and Supplementary Movie 3). Alternatively, some cells successfully completed division through asymmetric partitioning of vimentin, implying retention of vimentin(1-411) in one daughter cell and rejuvenation of the other by initial elimination of vimentin coils (Fig. 2a, lower panels, and Supplementary Movie 4). These characteristics are schematized in Fig. 2b. In a whole, the time that cells expressing vimentin(1-411) spend in mitosis is greatly lengthened compared to cells harboring vimentin wt (Fig. 2c). Moreover, the proportion of inherited vimentin is highly dissimilar in vimentin(1-411)-expressing cells that succeed to divide (Fig. 2d). Cytokinetic defects were also apparent, given the higher proportion of vimentin(1-411)-expressing cells showing aberrant or multiple nuclei (Fig. 2e). Additionally, we explored potential chromosome segregation defects associated with the presence of vimentin (1-411), and found that the frequency of lagging chromosomes in cells reassuming mitosis upon release of a nocodazole block was significantly higher in cells harboring vimentin(1-411) than in wt (Fig. 2f). Together, these results indicate that the tail domain is necessary for normal dynamics of vimentin filaments during mitosis, and this, in turn, is required for the adequate progression of cell division.

**Vimentin reaches the cell cortex in mitosis entailing actin.** Vimentin filaments maintain a close crosstalk with microtubules and microfilaments, which influence vimentin distribution. The interplay of vimentin wt or (1-411) with tubulin and actin was analyzed by disruption of these structures with various agents (Table 1). In SW13/cl.2 cells, transfected vimentin wt partially co-localized with tubulin in interphase, whereas in mitosis, vimentin underwent peripheral redistribution and tubulin concentrated at the mitotic spindle (Fig. 3a). Conversely, vimentin(1-411) bundles were largely unconnected to microtubules in resting cells, but remained adjacent to the mitotic spindle during cell division, frequently interfering with the position of chromosomes or microtubules (Fig. 3a). Acute microtubule disruption with nocodazole completely blocked the formation of the mitotic spindle and caused misorientation of chromosomes, without altering the peripheral distribution of vimentin wt in mitosis or the presence of vimentin(1-411) bundles (Fig. 3a).

Vimentin wt shows little coincidence with filamentous actin (f-actin) in interphase (Fig. 3b). In mitosis, f-actin accumulates at the cell periphery forming the actomyosin cortex, a stiff structure that allows spindle formation and orientation and maintains the cells spherical shape[34,35]. Interestingly, peripherally distributed vimentin appeared to line the internal surface of the actomyosin cortex, partially overlapping with actin (Fig. 3b, fluorescence intensity profiles). Conversely, vimentin(1-411) did not follow the actin pattern under any condition (Fig. 3b, lower panels). Importantly, vimentin cortical association was not a mere consequence of cell rounding, since newly plated round cells showed vimentin perinuclear distribution, clearly unrelated to actin (Supplementary Fig. 1a).

Although vimentin and actin show a reciprocal regulation at several cellular structures in resting cells[36,37], their interplay in mitosis has not been explored. Disruption of actin polymerization with cytochalasin B (Fig. 3c) elicited a patchy f-actin distribution without severely affecting vimentin wt in resting cells. In mitotic cells, vimentin lost its homogeneous cortical distribution and frequently appeared in bundles entangled with dividing chromosomes, reminiscent of vimentin(1-411), which was not further altered by cytochalasin B (Fig. 3c, lower panels). Latrunculin A markedly decreased f-actin, leading to scattered aggregates in interphase cells (Fig. 3d), and loss of cortical f-actin in mitosis, which correlated with vimentin bundling and intertwining with chromosomes. Jasplakinolide elicited actin aggregates co-localizing with vimentin filaments in resting cells, and a more irregular actin cortex in mitosis that partially retained vimentin structures (Fig. 3e). The effects of actin disrupting strategies are reflected in a decrease in the proportion of cortical vimentin at mid-cell height (Fig. 3f). Interestingly, treatment with C3 toxin to inhibit Rho proteins, which are important for actomyosin cortex assembly[38,39], or with the myosin ATPase inhibitor blebbistatin[40,41] also compromised vimentin peripheral distribution (Supplementary Fig. 1c).

Endogenous vimentin in Vero cells (Fig. 3g) and in parental SW13 cells (Supplementary Fig. 1d) showed similar responses to

**Table 1 Agents used to disrupt cytoskeletal structures**

| Agent | Target | Action | Effect on cortical vimentin |
|---|---|---|---|
| Nocodazole | Tubulin | Polymerization inhibition | No effect |
| Cytochalasin B | Actin | Polymerization inhibition | Dislodgement |
| Latrunculin A | Actin | Polymerization inhibition | Dislodgement |
| C3 toxin | RhoA⇒actin | Polymerization inhibition | Dislodgement |
| Jasplakinolide | Actin | Promotion and stabilization of f-actin | Partial dislodgement |
| Blebbistatin | Myosin | Inhibition of myosin II ATPase | Partial dislodgement |

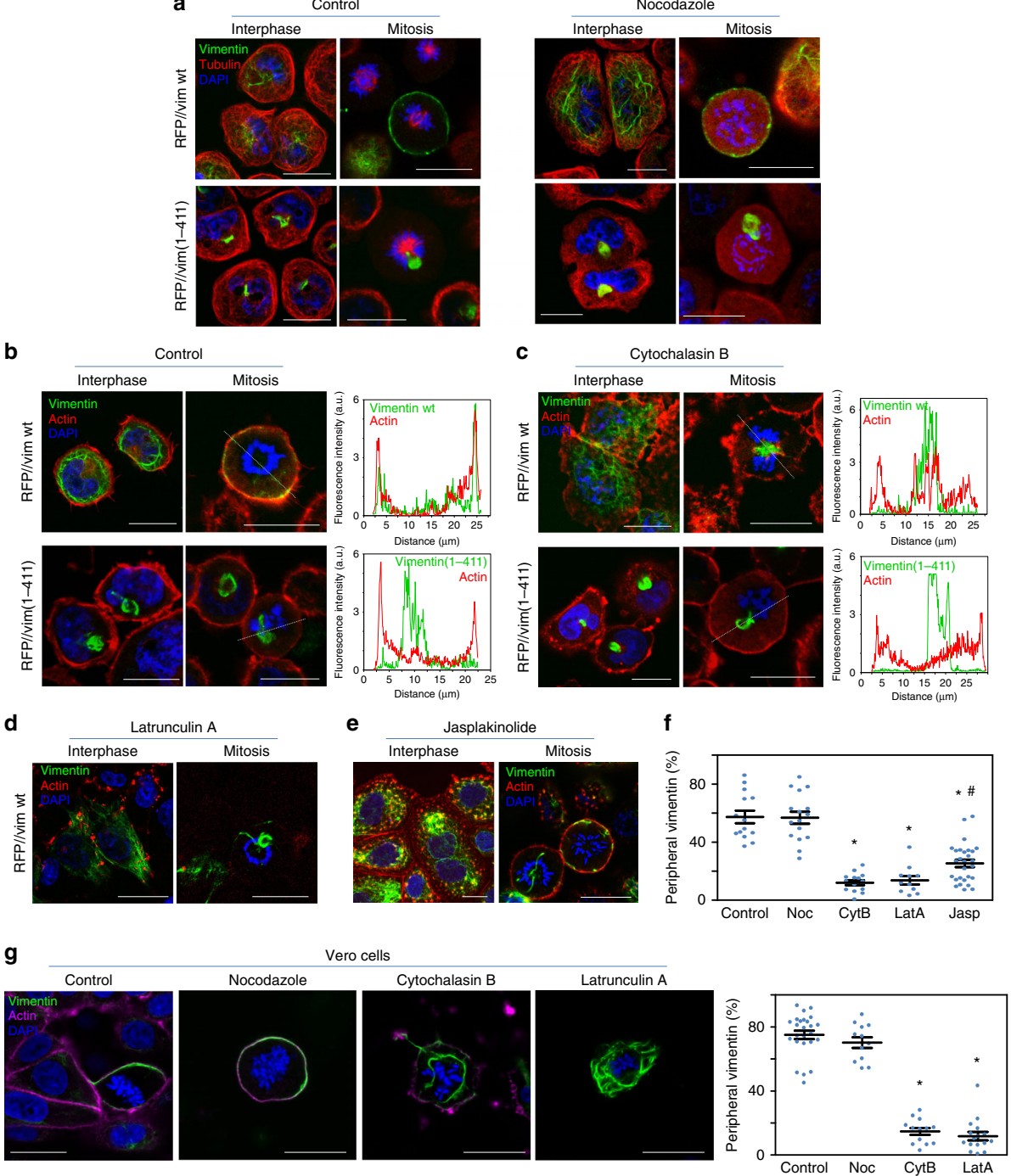

**Fig. 3** Effect of microtubule or actin filament disruption on the distribution of vimentin wt and vimentin(1-411). **a–d** The distribution of vimentin and tubulin or f-actin was assessed in interphase and mitotic cells. **a** SW13/cl.2 cells transfected with RFP//vimentin wt or (1-411) to express untagged vimentin proteins, were treated in the absence or presence of 5 μM nocodazole for 30 min in serum-free medium. Vimentin (green) and tubulin (red) were visualized by immunofluorescence. **b**, **c** SW13/cl.2 cells were transfected as above, cultured in the absence **b** or presence **c** of 10 μg/ml cytochalasin B for 30 min in serum-free medium. Vimentin (green) was visualized by immunofluorescence and f-actin was stained with Phalloidin (red). Fluorescence intensity profiles of vimentin and actin along the dotted lines are shown in the right panels. **d–f** SW13/cl.2 cells transfected with RFP//vimentin wt were treated in serum-free medium with 2.5 μM latrunculin A **d** or 50 nM jasplakinolide **e** for 30 min, and processed by immunofluorescence. Representative images of single sections taken at mid-height of interphase and dividing cells are shown. **f** Quantification of the proportion of vimentin located at the cell periphery upon disruption of microtubules or actin. Noc nocodazole; CytB cytochalasin B; LatA latrunculin A; Jasp jasplakinolide (*$p < 10^{-7}$ vs. control; #$p < 0.02$ vs. CytB and LatA). **h** Vero cells were treated with the indicated agents and the distribution of endogenous vimentin and actin in mitotic cells was assessed as above. Graph shows the proportion of vimentin associated with the cell periphery (*$p < 0.001$ vs. control). The number of determinations for the experimental conditions shown in graphs from left to right was the following: **f** 14, 16, 14, 11, 28; **g** 25, 12, 13, 16. Average values ± SEM are shown. All $p$-values were obtained with two-tailed, unpaired Student's $t$-test. Data are available in the Source Data file. Scale bars, 20 μm

cytoskeleton-disrupting agents, with conserved cortical localization in nocodazole-treated cells and marked dislodgement in response to cytochalasin B or latrunculin A. Together, these results support an important role of f-actin and cell cortex integrity in the tail-dependent mitotic redistribution of vimentin.

**Vimentin binds the mitotic cell cortex in several cell types**. The reorganization of vimentin in mitosis is cell type-dependent[42,43]

and responds to two main different patterns, consisting in filament disassembly or preservation. Therefore, we explored the association of vimentin with the actin cortex in cells of various origins, including primary cells and both cancer and non-cancer cell lines, of which, MCF7 are vimentin-deficient and the rest express endogenous vimentin (representative images are shown in Fig. 4a and fluorescence profiles in Supplementary Fig. 2). In MCF7 cells, transfected vimentin showed signs of disassembly in mitosis and limited cortical association. U-251 MG astrocytoma

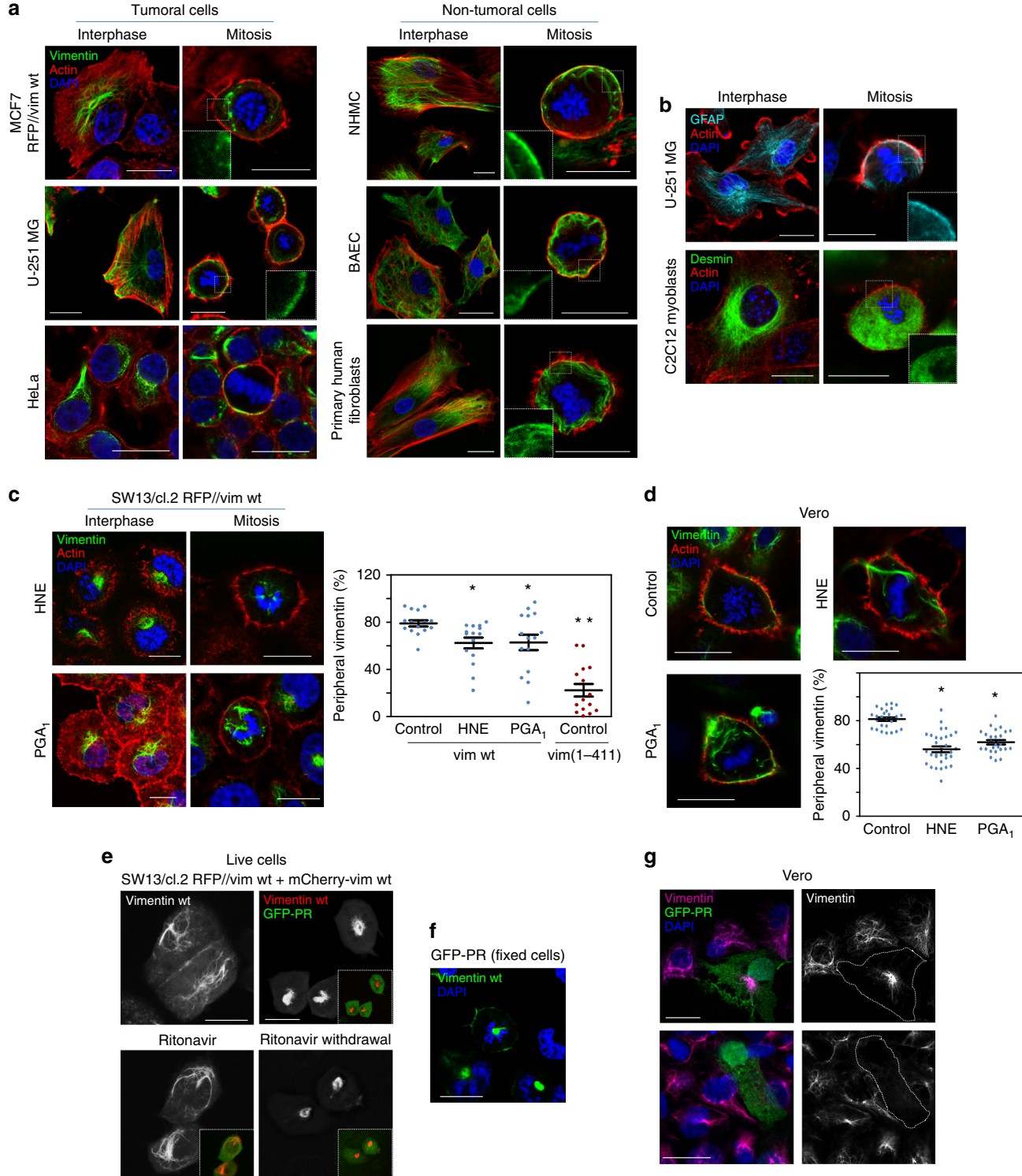

**Fig. 4** Cortical association of type-III intermediate filaments in several cell types and pathophysiological conditions. **a** The distribution of f-actin and vimentin in interphase and mitotic cells was assessed in several tumoral or non-tumoral cell types, the characteristics of which are specified in Supplementary Table 1. MCF7 cells were transfected, as indicated. Representative single merged sections at mid-cell height are shown. Individual channels are depicted in Supplementary Fig. 2. **b** The distribution of GFAP and desmin was assessed by immunofluorescence in mitotic human U-251 MG astrocytoma cells and undifferentiated murine C2C12 myoblasts, respectively. Insets (**a**, **b**) depict enlarged areas of the cell periphery showing vimentin (**a**) or GFAP and desmin (**b**) distribution. **c** SW13/cl.2 cells expressing untagged vimentin wt or **d** Vero cells were treated with electrophilic lipids, 10 μM 4-hydroxynonenal (HNE) for 4 h or 20 μM prostaglandin A$_1$ (PGA$_1$) for 20 h, and processed as above. Graphs show the proportion of peripheral vimentin. For comparison, values corresponding to vimentin(1-411) are included in the graph in **c**. *$p < 0.05$, **$p < 10^{-7}$ vs. control in **c** and *$p < 10^{-10}$ vs. control in **d**. The number of determinations for the experimental conditions shown in graphs from left to right was the following: **c** 15 for all conditions; **d** 26, 32, 26. Average values ± SEM are shown. All $p$-values were obtained with two-tailed, unpaired Student's $t$-test. Source Data file contains original datasets. **e** SW13/cl.2 cells were transfected with a combination of RFP//vimentin wt (80%) plus a tracer amount of mCherry-vimentin wt (20%), to monitor vimentin filaments (grayscale in main images, red in insets) in live cells. Additionally, cells were transfected with the HIV-type I protease (GFP-PR, green fluorescence depicted in insets). In upper panels, cells were imaged 24 h after transfection. In the lower panels, the HIV inhibitor ritonavir was added immediately after transfection and cells were imaged 24 h later (left panel). Subsequently, ritonavir was removed and cells were imaged 5 h later (lower right panel). **f** Lastly, cells were fixed and stained with DAPI to identify mitotic cells. Vimentin is artificially colored in green. **g** Vero cells were transfected with GFP-PR (green) and 24 h later, vimentin was detected by immunofluorescence (magenta). Left images show overlay projections, right images depict vimentin distribution in gray scale. Results are representative from three experiments with similar effects. Scale bars, 20 μm

cells displayed a heterogeneous pattern, with apparent cortical localization in some cells (Fig. 4a). In contrast, in HeLa (Fig. 4a), and SW13 parental cells (Supplementary Fig. 1d), filamentous structures of endogenous vimentin adopted a preferential peripheral distribution, close to the actin cortex. An intimate cortical association was also evident in primary human mesangial cells (NHMC, Fig. 4a), whereas in primary human fibroblasts and bovine aortic endothelial cells (BAEC) vimentin showed points of contact with the actin cortex but the filamentous structures extended more into the cytoplasm (Fig. 4a and Supplementary Fig. 2). Therefore, the extent of cortical association of vimentin filaments in mitosis is cell-type dependent.

Importantly, filaments of glial fibrillary acidic protein (GFAP), another type III intermediate filament protein, also adopted a peripheral distribution in mitotic U-251 MG astrocytoma cells, whereas desmin filaments disassembled in mitotic undifferentiated C2C12 myoblasts, showing a predominantly diffuse cytoplasmic staining (Fig. 4b).

**Pathophysiological agents alter vimentin cortical relocation.** We then assessed the distribution of vimentin filaments in the presence of pathophysiological agents known to cause vimentin collapse. The inflammatory lipid mediators 4-hydroxynonenal (HNE) and prostaglandin A$_1$ (PGA$_1$) induced vimentin juxtanuclear condensation in interphase SW13/cl.2 cells expressing vimentin wt (Fig. 4c), as previously observed[11]. Moreover, they significantly dislodged vimentin from the actomyosin cortex in mitotic cells (Fig. 4c), according to the peripheral/total vimentin fluorescence ratio, which is shown in comparison with that of vimentin(1-411) (Fig. 4c, graph). Mitotic Vero cells treated with HNE or PGA$_1$ also showed more abundant cytoplasmic vimentin bundles and a lower proportion of peripheral vimentin (Fig. 4d). The HIV-type I protease has been reported to cleave vimentin tail leading to abnormal filament distribution[44]. Expression of a HIV-protease construct (GFP-PR) in vimentin wt-transfected SW13/cl.2 cells was associated with collapse of vimentin filaments into curly juxtanuclear bundles (Fig. 4e), reminiscent of the vimentin (1-411) distribution. Ritonavir, a reversible HIV-protease inhibitor, blocked vimentin collapse, whereas its withdrawal allowed fast vimentin condensation, indicating a role for protease activity (Fig. 4e). HIV-protease-induced vimentin accumulations remained in mitosis and concentrated close to the dividing chromosomes (Fig. 4f). Notably, Vero cells transfected with GFP-PR showed either vimentin condensation or loss of immunoreactivity (Fig. 4g). Nevertheless, no mitosis of GFP-PR-positive

Vero cells could be observed, suggesting interference of GFP-PR with cell division. Together, these results show that various pathophysiological agents cause anomalous vimentin distribution, hampering cortical localization in mitosis.

**Intimate cortical vimentin–actin intertwining in mitosis.** Further insight into the interaction of vimentin with the actomyosin cortex during cell division was obtained by superresolution microscopy (STED). Mitotic cells were identified by differential interference contrast (DIC), as round cells lacking delimited nuclei and showing a central rugged area coincident with the position of chromosomes (as exemplified by DAPI staining and confocal microscopy in Fig. 5a). STED microscopy showed vimentin filaments next to the actin cortex both in Vero cells, where the colocalization mask unveils the intimate contact between endogenous vimentin and cortical actin (Fig. 5b), and in vimentin-transfected SW13/cl.2 cells (Fig. 5c), where vimentin filaments intermingle with actin at some points (Fig. 5c, upper two rows) or run between two actin layers (Fig. 5c, lower two rows). Single section analysis of actin–vimentin co-localization showed a closer connection at certain locations along the actin cortex, evidenced by the colocalization mask (Fig. 5c), suggesting the existence of docking or penetration sites of vimentin in this structure. Interestingly, the proportion of f-actin showing cortical localization in these sections was 45.7%, whereas the proportion of cortical vimentin was 57% (Fig. 5d), for which, overlap and Pearson coefficients of colocalization with actin were 0.92 and 0.56, respectively (Fig. 5e). Three-dimensional reconstructions revealed a robust basket-shaped framework of vimentin filaments of diverse orientations (Fig. 5f). Actin formed a hollow sphere constituted by elongated patches or bundles mostly oriented perpendicularly to the support surface, as illustrated in the 3D-reconstruction of the cell bottom half (Fig. 5g and Supplementary Movie 5). Interestingly, this reconstruction evidences points of vimentin filament protrusion through the actin cortex (Fig. 5g, arrow).

**Vimentin affects cortical actin organization in mitosis.** A more comprehensive analysis of the vimentin–actin interaction at the cortex of mitotic cells was carried out as schematized in Fig. 6a. First, we obtained 3D-reconstructions of the cortex of vimentin-positive and vimentin-negative cells. This confirmed that some cells displayed ample segments of vimentin at the external surface of the cortex, interwoven with f-actin structures, which in mitotic SW13/cl.2 cells appear mainly as elongated bundles perpendicular

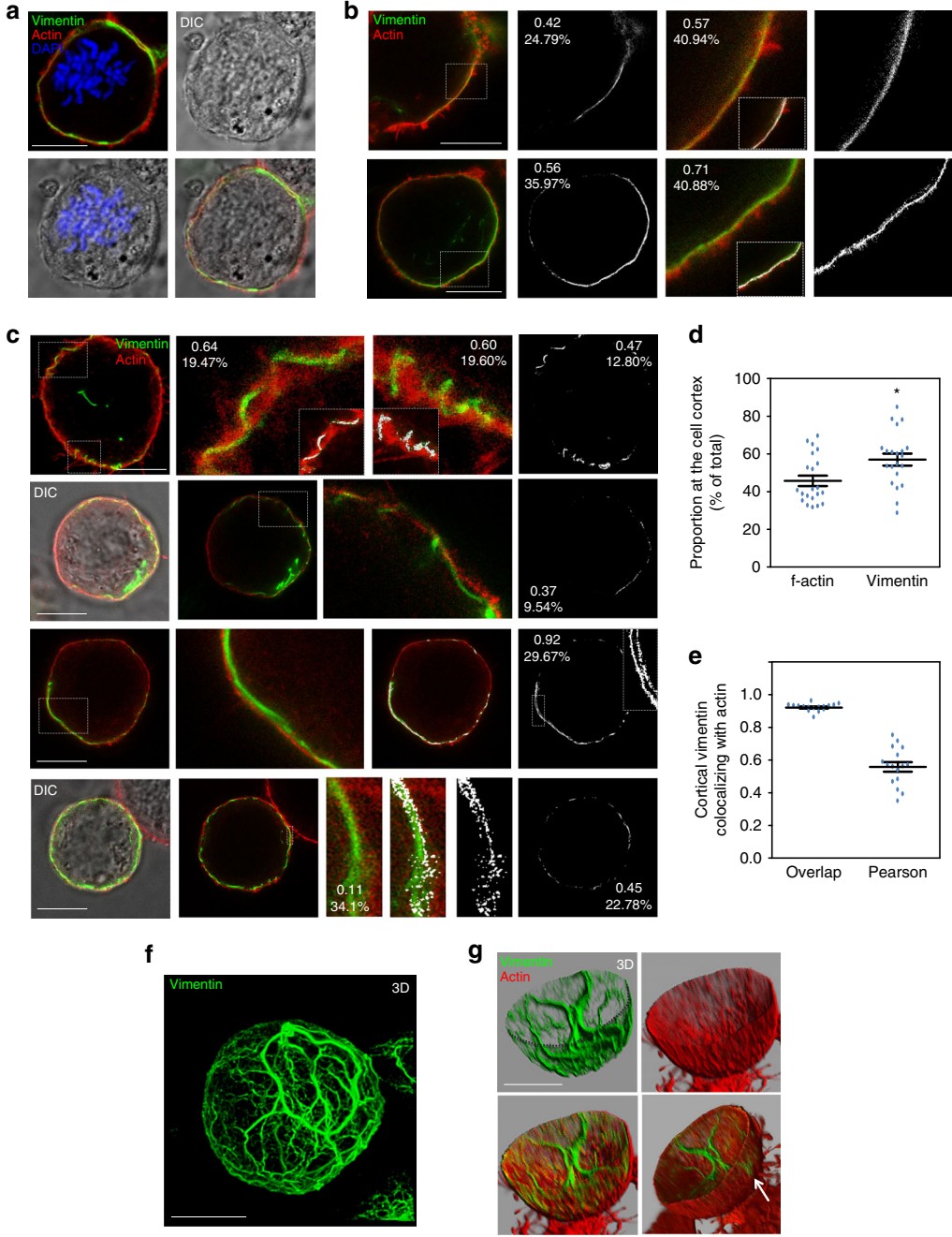

**Fig. 5** Analysis of the relative positions of vimentin and actin in mitosis by STED superresolution microscopy. **a** Confocal microscopy images illustrating the identification of mitotic Vero cells by the typical aspect of dividing chromosomes in DIC, confirmed by DAPI staining in merged images. The distribution of f-actin and vimentin was monitored by TRITC-phalloidin staining and immunofluorescence with Alexa488-conjugated V9 antibody, respectively. **b** Mitotic Vero cells, identified by DIC visualization, were analyzed by STED and images of several cells are shown. Colocalization masks (shown in white) or enlargements of areas delimited by dashed rectangles are shown to the right. Co-localization analysis was performed with Leica software. Numbers in insets represent the Pearson's coefficient and the percentage of co-localization for the regions shown. **c** SW13/cl.2 cells stably transfected with RFP// vimentin wt were treated with 0.4 µM nocodazole overnight, to increase the proportion of mitotic cells. Vimentin and f-actin were detected as above. STED images of several cells are shown. Overlays with DIC images are confocal images. Colocalization masks (white) or enlargements of areas delimited by dashed rectangles are shown to the right. Numbers in images represent the Pearson's coefficient and the percentage of co-localization, respectively, for the whole cell or for the regions enlarged. **d** Proportion of f-actin or vimentin located at the cell periphery ($n = 21$, $*p < 0.02$ by two-tailed, unpaired Student's $t$-test). **e** Colocalization of cortical vimentin with f-actin analyzed by calculation of overlap and Pearson coefficients for the cortical ROI ($n = 16$). Datasets for **d** and **e** are provided in the Source Data file. Average values ± SEM are shown. **f** 3D-reconstruction of vimentin organization, after deconvolution of the green channel using Imaris software, for one representative cell treated as in **c**. **g** 3D-reconstruction using the basal half of the sections from the same cell in order to show the inside and the outside of the sphere. Single channels (upper panels) and merged images (lower panels) are shown. The semi-sphere edge is marked in the green channel (dotted line). The bottom-right image is a snapshot of Supplementary Movie 5. The arrow points at a protrusion of vimentin through the actin cortex. Scale bars, 10 µm

to the substrate (Fig. 6b and Supplementary Movies 6 and 7). Next, 2D-map projections from the image stacks were prepared for global visualization and quantitation of the cortex[45] (Fig. 6c). Fluorescence intensity profiles of these projections illustrated the alternate distribution of actin and vimentin signals at some points (Fig. 6d). 2D-maps from vimentin-negative cells (non-transfected or transfected with the RFP// empty vector) presented higher standard deviation of f-actin pixel brightness, indicating wider variations in f-actin distribution (Fig. 6e). Additionally, we analyzed orthogonal projections of vimentin-positive and vimentin-negative mitotic cells (Fig. 6f). Notably, vimentin filaments could be detected both at the top and bottom of vimentin-positive cells (Fig. 6f, arrowheads), with a particular enrichment of robust lattices at the basal layer, next to the substrate. These structures were obviously absent from non-transfected cells, but also from cells expressing vimentin(1-411), which was frequently retained close to a cytoplasmic ring-shaped accumulation of f-actin that was clearly detectable in some cells (Fig. 6f, inset, arrow). Whether this structure is related to the inner actin ring, which has been involved in spindle positioning[46] requires further study. The interplay between the distributions of f-actin and vimentin was reflected by a higher proportion of basal vimentin in cells expressing vimentin wt, together with a lower proportion of f-actin at the basal region of these cells (quantitated in Fig. 6g). Additionally, the basal vimentin wt lattice was associated with a lower intensity of f-actin signal, in absolute values, and a lower standard deviation of pixel brightness at this location, suggestive of less polymerized actin structures (Fig. 6h). Altogether, these results position vimentin as an important player at the actin cortex of mitotic cells, exerting a measurable impact on its characteristics, particularly on the abundance and distribution of f-actin, which could influence cell division dynamics.

**Serial vimentin truncations gradually impair mitosis.** Structural determinants allowing vimentin to reach the cell periphery in mitosis were explored by analyzing the distribution of mutants bearing several C-terminal deletions (Fig. 7a). A vimentin(1-423) truncated form, which mimics the reported HIV-protease cleavage product[44,47,48], formed curly bundles in the nuclear vicinity (Fig. 7b), similar to vimentin(1-411). Vimentin(1-423) coiled bundles mainly remained near the condensed chromosomes in mitosis, either interfering with the mitotic spindle or locating at one of the poles (Fig. 7c). Often, multi-nucleated cells containing coiled vimentin(1-423), and in some cases DNA, in the space between nuclei were found (Supplementary Fig. 3), suggesting cytokinetic defects. Time-lapse monitoring of cells expressing vimentin(1-423) confirmed marked mitotic alterations, including vimentin asymmetric partitioning, delayed mitosis and cell death (Fig. 7d and Supplementary Movies 8 and 9). Moreover, vimentin (1-423), was often retained close to cytoplasmic accumulations of f-actin in the proximity of the spindle, and did not reach cortical actin (Fig. 7e).

Vimentin(1-448) (Fig. 7a), yielded a heterogeneous pattern with both extended filaments and robust bundles or accumulations (Fig. 7f). These persisted in mitotic cells, sometimes appearing at basal planes or at the cell periphery (Fig. 7g). Mitotic cells frequently exhibited delayed separation ending in cell death, indicative of cytokinetic failure, or completed mitosis through vimentin asymmetric partition (Fig. 7h and Supplementary Movies 10 and 11). Nevertheless, some peripheral filamentous vimentin could be detected, lying adjacent to the actomyosin cortex (Fig. 7i).

Finally, a construct with a shorter C-terminal deletion, vimentin(1-459), formed filaments similar in morphology and extension to those of vimentin wt, although ~18% of the cells also showed small bundles or curls (Fig. 7j). In mitosis, vimentin(1-459) adopted a mainly peripheral distribution, although some cells presented filaments intertwined with chromosomes (Fig. 7k). In time-lapse monitoring (Fig. 7l and Supplementary Movies 12 and 13), cells lacking vimentin bundles underwent basically normal mitosis with even vimentin distribution between daughter cells. Conversely, cells harboring vimentin bundles showed a partial asymmetric distribution, with one daughter cell receiving most of vimentin(1-459) (Fig. 7l). Vimentin(1-459) coincided with some segments of the actomyosin cortex, whereas some filaments persisted in the central area (Fig. 7m). Thus, deletion of the last seven amino acids induces a mild perturbation of vimentin distribution in mitosis. Panels (n–o) recapitulate these observations showing the correlation between the length of the remaining tail segment and the ability of each truncated vimentin to form an extended network in interphase cells (measured as the cellular area covered by every vimentin construct) (Fig. 7n). Consistent with the deleterious effects of tail truncations, the severity of the truncation correlates with the proportion of cells showing vimentin accumulations close to the diving chromosomes in mitosis (Fig. 7o). In turn, this proportion directly correlates with the frequency of mitotic defects, assessed as the duration of mitosis (Fig. 7o, red labels), and the presence of cells with multiple or aberrant nuclei (Fig. 7o, blue labels). Nevertheless, additional deleterious effects of the tail-truncated constructs, including potential interference with signal transduction pathways cannot be excluded.

Taken together, these results reveal that step-wise deletion of the tail gradually impairs normal vimentin assembly in cells and redistribution in mitosis, with vimentin truncation at L423 or I411 abolishing mitotic peripheral localization in association with the more severe mitotic defects.

**Vimentin particles associate with the cell cortex in mitosis.** To discard that failure to associate with the cell cortex in mitosis could be due to intense bundling, we used GFP-vimentin fusion constructs, which do not form full filaments in SW13/cl.2 cells[11,49]. First, the organization of GFP-vimentin wt and all the truncated variants was studied (Fig. 8a, b). GFP-vimentin wt formed a uniform lattice of squiggles or short filaments[11] (Fig. 8a). Conversely, GFP-vimentin(1-411) could not reach the squiggle stage and formed only bright dots. GFP-vimentin(1-423) exhibited a mixed pattern consisting of dots, small swirls and occasional short filaments. Longer constructs, namely, GFP-vimentin(1-448) and GFP-vimentin(1-459) often presented short filaments or squiggles (Fig. 8a). Thus, GFP-fusion constructs displayed a gradual sequence-dependent impairment of particle elongation in vimentin-deficient cells, stressing that even small truncations of the tail domain have a detectable impact (quantitated in Fig. 8a, graph).

To increase the proportion of mitotic cells, we employed a mild nocodazole treatment. As nocodazole can affect vimentin distribution, we chose the minimum concentration to attain conditions under which actin or vimentin were not appreciably altered (see the "Methods" section and Supplementary Fig. 1b). In cells arrested in mitosis by mild nocodazole treatment, GFP-vimentin constructs frequently showed a diffuse background, suggestive of a higher extent of disassembly than untagged vimentin. Nevertheless, GFP-vimentin wt structures were clearly detected at the cell cortex in mitosis co-localizing with actin (Fig. 8c). In sharp contrast, GFP-vimentin(1-411) dots appeared scattered throughout the cell. This lack of cortical association cannot be solely attributed to defective elongation, since dots formed by full-length GFP-vimentin C328S, which is also elongation-incompetent[11], clearly relocated to the periphery of

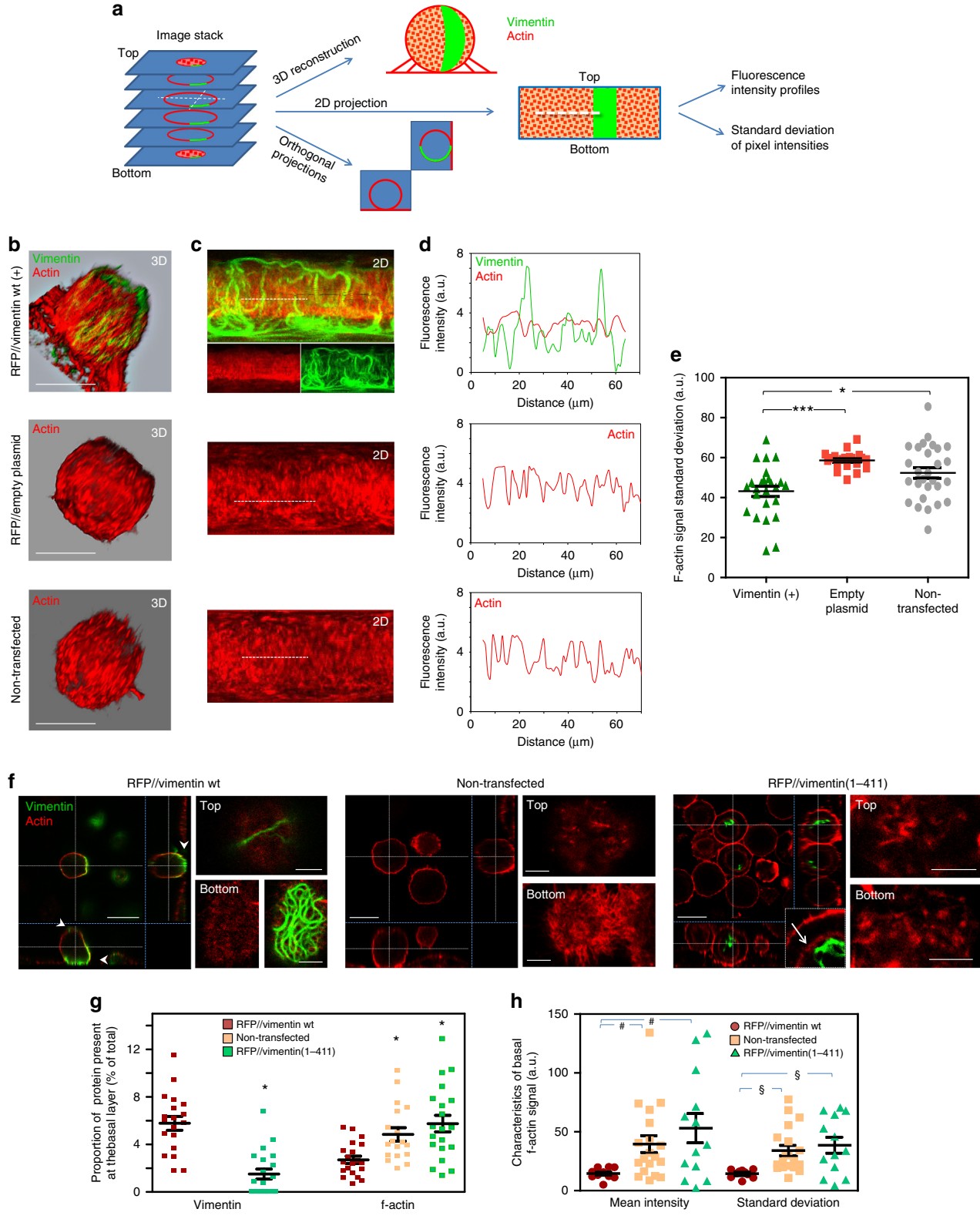

mitotic cells (Fig. 8c). GFP-vimentin(1-423) accumulations also failed to associate with the actin cortex, frequently appearing near the cytoplasmic f-actin ring, whereas >60% of GFP-vimentin(1-448) dots redistributed to the cell periphery and cortical localization of GFP-vimentin(1-459) mixed structures was preserved (quantitated in Fig. 8d). Notably, the 448-459 segment contains the RDG motif, amino acids 450-452 (Fig. 8b), which is

important for vimentin assembly in vitro and in cells[50,51]. To assess the importance of this motif we studied the impact of a G452V mutation, which strongly interferes with normal assembly in vitro[50]. Interestingly, although GFP-vimentin G452V only formed bright dots, indicating defective assembly, they efficiently associated with the cell cortex in mitosis (Fig. 8c, d). Additionally, we quantitated the proportion of GFP-vimentin constructs

**Fig. 6** Vimentin expression affects the properties of the actin cortex in mitosis. **a** Scheme of the different approaches employed to analyze mitotic cell properties. XY image stacks were used to obtain 3D-reconstructions of mitotic cells, 2D-cartographic, and orthogonal projections. **b** 3D-reconstruction of SW13/cl.2 cells, stably expressing vimentin wt (top), RFP from the bicistronic pIRES DsRed Express2 empty plasmid (RFP// empty plasmid, middle), or non-transfected (bottom), scale bars, 20 μm. **c** 2D-maps from the cells shown in (b). For the vimentin-expressing cell, the merged and single channels are shown. **d** Fluorescence intensity profiles along the white dotted lines drawn on the 2D-maps. **e** Standard deviation values of the f-actin signal of 2D-maps (*$p < 0.05$, ***$p < 0.001$). **f** SW13/cl.2 cells were transfected with RFP//vimentin wt (untagged vimentin wt) or RFP//vimentin(1-411) (untagged tailless vimentin), or not transfected (non-transfected). Orthogonal projections illustrate the positions of the vimentin constructs. Arrowheads mark the appearance of vimentin wt at the top and at the bottom of the cell. The inset shows an enlarged image illustrating the localization of vimentin(1-411) with respect to the cytoplasmic f-actin ring (arrow). Right panels show the top and bottom sections for every construct. Scale bars in right panels, 5 μm. **g** Proportion of vimentin or f-actin present at the basal layer with respect to the total cellular content (*$p < 0.01$ vs. cells transfected with RFP//vimentin wt). **h** Graph summarizing the mean and standard deviation of the f-actin signal intensity at the bottom section of the cell (#$p < 0.02$ and §$p < 0.01$ vs. cells transfected with RFP//vimentin wt). The number of determinations for the experimental conditions shown in graphs from left to right was the following: **e** 25, 20, 28; **g** 19, 20, 20, 18, 20; **h** 10, 24, 13, 10, 25, 13. Average values ± SEM are shown. *p*-values were obtained with ANOVA followed by Tukey's multiple comparison test in **e** an with the two-tailed, unpaired Student's *t*-test in **g** and **h**. Data are provided in the Source Data file

located at the cell periphery. This clearly confirmed the marked impact of tail truncations on mitotic cortical association, highlighting the importance of the segment 424–448 for this redistribution (Fig. 8e). Moreover, full filament elongation or correct filament assembly are not necessary for cortical association, since vimentin particles formed by constructs retaining all or most of the tail domain, as well as the G452V mutant, effectively relocate to the cell periphery. Nevertheless, a certain degree of organization seems necessary since constructs yielding mostly soluble, diffuse vimentin, including a GFP fusion construct of the tail domain, GFP-vim(412–466), and the assembly-incompetent vimentin Δ(3-74) mutant were not able to redistribute to the cell cortex in mitotic cells (Supplementary Fig. 4).

Thus, the tail domain is essential, but not sufficient, for vimentin cortical association in mitosis, and other structural or conformational factors appear necessary.

## Discussion

Vimentin plays critical functions in cell mechanics[52]. Nevertheless, its role in mitosis is not fully understood. Here we unveil the robust network formed by vimentin filaments in mitosis in several cell types. This framework intimately interacts with the actomyosin cortex, intertwining with actin, and affecting its properties. Several functions can be envisaged for this arrangement: to yield space for mitotic spindle organization, and, potentially, to modulate the robustness or stiffness of the actomyosin cortex in dividing cells. Therefore, these results warrant the pertinence to study vimentin, and other intermediate filaments, as players in the dynamics of the cell cortex in mitosis.

Vimentin organization in mitosis is cell-type dependent and responds to two main patterns: formation of a filament cage surrounding the mitotic spindle, or disassembly. Vimentin disassembly is accepted to be mediated by phosphorylation of N-terminal residues in combination with protein-protein interactions, reportedly, copolymerization with nestin[42,43,53]. More than 11 residues at the N-terminus of vimentin have been shown to be phosphorylated in mitosis by various kinases in a coordinated manner[43]. Curiously, although phosphorylation of vimentin tail residues has also been reported, they do not seem to play a role in mitotic disassembly[17]. Under our conditions, vimentin filament disassembly or persistence in mitosis could be related to nestin levels, as previously demonstrated[17]. Accordingly, dividing SW13/cl.2 cells, which are nestin-negative, show persisting filaments, nestin-positive MCF7 cells[54] display vimentin disassembly, and other cell types expressing variable nestin:vimentin proportions, such as U-251 MG astrocytoma cells[33] or BAEC[55], show mixed patterns.

Our results indicate that, if not disassembled, vimentin filaments should undergo mitotic cortical translocation or anchorage to facilitate mitosis progression. This is substantiated by the striking behavior of vimentin(1-411), which does not reach the actomyosin cortex and interferes with the mitotic apparatus causing aberrant mitosis. Moreover, a correlation appears to exist between the extent of vimentin tail truncation, the interference with chromosomes and the perturbation of mitosis. This raises potential cytotoxic implications of vimentin cortical dislodgement in pathophysiological settings, as observed upon lipoxidation or HIV-protease action, although damage or cleavage of macromolecules different than vimentin could contribute to these effects.

The intrinsically disordered C-terminal vimentin domain has been proposed to undergo conformational rearrangements during filament assembly and to participate in protein-protein interactions, including actin[56,57]. Our results indicate that in mitosis, filamentous vimentin interacts with the actomyosin cortex showing points of co-localization with actin. However, this interaction could take place through other proteins, including the scaffold protein plectin, chaperones or actin-associated proteins[58]. Additionally, vimentin protrudes through the actomyosin cortex at some points, for which sites of attachment at the plasma membrane involving protein receptors or lipid domains, cannot be excluded.

A complex interplay between actin and vimentin at several organization levels exists in resting cells[36]. Actin limits transport of vimentin ULF along microtubules[59], and actomyosin arcs interact with vimentin filaments through plectin and drive their retrograde movement, thus promoting vimentin perinuclear localization[37]. In turn, vimentin restricts retrograde movement of the arcs and restrains actin polymerization and stress fiber assembly[60]. Nevertheless, the actin-vimentin interaction in mitosis has not been addressed to our knowledge. The actomyosin cortex provides tension, which together with osmotic pressure controls cell rounding. Actin organization is a key factor in contractile tension generation[61]. Importantly, our results clearly show that integrity of the actomyosin cortex is necessary for vimentin cortical association, but also unveil an impact of vimentin on cortical actin organization. In mitotic SW13/cl.2 cells, the lower dispersion of f-actin signal in global cortex 2D-projections and the lower intensity and proportion of f-actin at basal planes in vimentin-positive cells indicate that vimentin influences actin distribution exerting a negative feedback on the formation of defined f-actin structures. Thus, our studies open the way for dissecting the consequences of vimentin–actin interplay on actomyosin contractility or stiffness in mitosis, and its potential implications in cell rounding or spindle orientation[62].

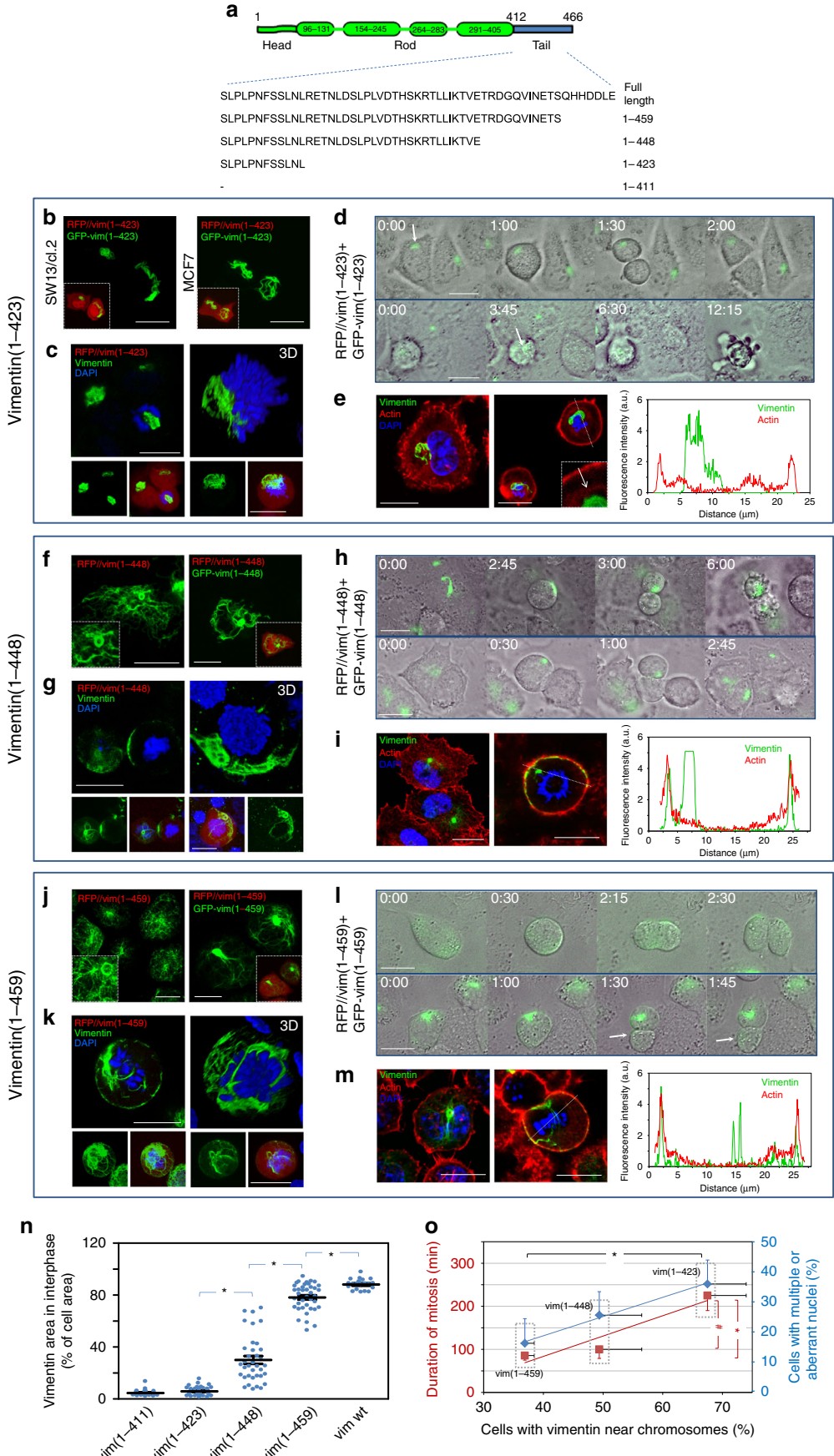

**Fig. 7** Organization and mitotic distribution of several C-terminal truncated vimentin mutants. **a** Scheme of truncated vimentin mutants. **b** Overall projections of live SW13/cl.2 or MCF7 cells transfected with RFP//vimentin(1-423) plus GFP-vimentin(1-423). Insets, merged channels showing RFP fluorescence to delimit the cell contour. **c** Immunofluorescence of SW13/cl.2 cells transfected with RFP//vimentin(1-423), illustrating truncated vimentin distribution in mitosis; left image, single section at mid-cell height; right image, 3D-projection; lower panels, overall projections of vimentin alone (green) or overlays of vimentin, DAPI and RFP fluorescence. **d** Live cells transfected with RFP//vimentin(1-423) plus GFP-vimentin(1-423) monitored by time-lapse microscopy, illustrating asymmetric partition (upper panels) and mitotic catastrophe (lower panels). **e** Cells were transfected as in **c** and vimentin and f-actin distributions monitored in interphase (left image) and mitosis (right image; arrow, cytoplasmic f-actin); right panel, vimentin and f-actin fluorescence intensity along the dotted line in the mitotic cell. **f** and **j** SW13/cl.2 cells were transfected with RFP//vimentin(1-448) in **f**, or RFP//vimentin(1-459) in **j**, alone or alongside the corresponding GFP-fusion construct, as indicated; left panels, vimentin immunofluorescence; right panels, vimentin visualization in live cells. **g** and **k** Cells transfected with RFP//vimentin(1-448) in **g**, or RFP//vimentin(1-459) in **k** were analyzed as in **c**. **h** Time-lapse monitoring of RFP//vimentin(1-448)-transfected cells illustrating mitotic failure (upper sequence) and asymmetric division (lower sequence). **l** Monitorization of RFP//vimentin(1-459)-transfected cells undergoing normal (upper panel) or asymmetric division (lower panel; arrow, vimentin vestige). **i** and **m** Vimentin and f-actin distribution in resting and mitotic RFP//vimentin(1-448)- (**i**) or RFP//vimentin(1-459)-(**m**) transfected cells, as described for **e**. **n** Ability of vimentin constructs to form an extended network in interphase, quantitated as percentage of cell area covered by vimentin (*$p < 0.001$). **o** Correlation between the proportion of cells with vimentin near dividing chromosomes ($x$-axis) for each vimentin-truncated mutant and the corresponding mitosis duration (left $y$-axis, red labels, $y = 4.7251x - 105.33$; $R^2 = 0.899$) or the proportion of cells showing multiple/aberrant nuclei (right $y$-axis, blue labels, $y = 4.4633x - 47.031$; $R^2 = 0.994$) (*$p < 0.01$; #$p < 0.02$). The number of determinations for the experimental conditions shown in graphs from left to right was the following: **n** 22, 26, 36, 39, 20 **o** duration of mitosis, 17, 15, 30; vimentin near chromosomes, three determinations totaling 51, 55, and 51 cells; cells with multiple or aberrant nuclei, three determinations totaling 155, 206, and 151 cells. Average values ± SEM are shown. All $p$ values were obtained by two-tailed, unpaired Student's $t$-test. Scale bars, 20 μm. Data are available in the Source Data file

Vimentin tail integrity is determinant for cortical association. On one hand, both untagged and GFP-fusion constructs show a graded impairment of cortical association upon serial tail truncations, the strongest impact occurring after deletion of the 43 distal residues. Therefore, the sequence comprised between residues 424–448, appears to contain important determinants for mitotic redistribution. This segment approximately coincides with a putative loop proposed to protrude from filaments and participate in protein–protein interactions[63,64]. On the other hand, it is clear that vimentin polymerization into full filaments is not necessary for cortical localization, since various constructs showing defective assembly or elongation, including GFP-vimentin C328S and GFP-vimentin G452V, effectively associate with the cell periphery. Nevertheless, the fact that the isolated tail domain or an N-terminal-altered mutant appear as diffuse proteins that do not undergo cortical association, suggests that contribution from several domains of vimentin could be necessary, potentially to achieve a certain level of organization and/or to adopt a precise conformation.

Thus, our results, summarized in Fig. 9, show that vimentin filaments redistribute to the cell periphery in mitosis in a tail domain-dependent manner. This reorganization implies a close interplay with the actomyosin cortex, with implications for intermediate filament dynamics during cell division, and opens the way for the search of strategies modulating these interactions.

## Methods

**Reagents**. Restriction enzymes and buffers were from Promega. Anti-vimentin antibodies were: mouse monoclonal V9 clone (sc-6260) and its Alexa-488 conjugate from Santa Cruz Biotechnology, and mouse anti-vimentin monoclonal antibody (V5255) from Sigma. Anti-actin (A2066) and anti-Hsp70 (H5147) were from Sigma, and anti-α-tubulin (ab52866) and anti-desmin (ab15200-1) from Abcam. Anti-GFAP (Z0334) was from Dako. C3 transferase toxin was from Cytoskeleton. Latrunculin A and jasplakinolide were from Santa Cruz Biotechnology. HNE and PGA₁ were from Cayman Chemical. 4,6-Diamidino-2-phenylindole (DAPI), cytochalasin B, nocodazole, blebbistatin, and ritonavir were from Sigma.

**Cell culture and treatments**. A list of the cell types used in the study is provided in Supplementary Table 1. SW13/cl.2 human adrenocarcinoma vimentin-deficient cells were the generous gift of Dr. A. Sarriá (University of Zaragoza, Spain)[31]. SW13/cl.2 stably transfected with the bicistronic plasmid RFP//vimentin wt (see below) have been generated in our laboratory[11]. Moderate levels of vimentin expression in this cell line were ensured by comparison with those present in human primary fibroblasts and Vero cells by western blot (Supplementary Fig. 5a).

SW13 parental cells (mixture of vimentin-positive and negative cells) from The European Collection of Authenticated Cell Cultures (ECACC 87031801) were acquired from Sigma. HeLa (CCL-2™), MCF7 (HTB-22™) human breast carcinoma cells and U-251 MG human glioblastoma astrocytoma cells (formerly known as U-373 MG, HTB-17™) were originally from ATCC and were authenticated by microsatellite amplification (short tandem repeat (STR)-PCR profiling), (Secugen, S.L., Madrid, Spain). Fibroblast-like African green monkey kidney cells (Vero) and C2C12 murine myoblasts were from the collection of Centro de Investigaciones Biológicas (Madrid). They were cultured in DMEM with 10% (v/v) fetal bovine serum (FBS) (Sigma or Biowest) and antibiotics (100 U/ml penicillin and 100 μg/ml streptomycin, Invitrogen). BAEC (BW-6001) were from Lonza and human mesangial cells (NHMC, CC-2559) were from Clonetics, and were cultured in RPMI1640 supplemented with 10% (v/v) newborn calf serum (Gibco) or FBS, respectively, and antibiotics. Primary human dermal fibroblasts from an adult donor (ref. AG10803) were obtained from the NIA Aging Cell Repository at the Coriell Institute for Medical Research (Camden, NJ). HAP1 vimentin-deficient cells (HZGHC003297c010) were from Horizon Genomics, GmbH (Vienna, Austria) and were cultured in DMEM F12 with FBS and antibiotics. All cell types were periodically confirmed to be free of mycoplasma contamination. Unless otherwise stated, treatments were carried out in serum-free medium. For acute microtubule disruption, cells were treated with 5 μM nocodazole for 30 min, whereas for mitotic arrest, cells were cultured in the presence of 0.4 μM nocodazole for 20 h in complete medium. This treatment was employed, when indicated, to increase the proportion of mitotic cells in conditions under which neither actin, nor vimentin organization were altered with respect to untreated cells and typical actin structures were preserved (Supplementary Fig. 1b). Disruption of f-actin was achieved by treatment with 10 μM cytochalasin B or 2.5 μM latrunculin A for 30 min, or 2 μg/ml C3 toxin for 3.5 h. Jasplakinolide was employed at 50 nM for 30 min. Blebbistatin was used at 20 μM for 1 h. For treatment with electrophilic lipids, cells were incubated in the presence of 10 μM HNE for 4 h or 20 μM PGA₁ for 20 h. Inhibition of transfected HIV protease was achieved by addition of 10 μM ritonavir in serum-containing medium, immediately after transfection. For removal of the inhibitor, cells were washed three times with fresh medium with serum, without antibiotics. For assessment of lagging chromosomes, cells transfected with RFP//vimentin wt or (1-411) were treated with 0.4 μM nocodazole to induce mitotic arrest, after which, cells were washed twice with complete medium for nocodazole removal and further incubated in fresh medium for 100 min to increase the proportion of cells in anaphase. Cells were subsequently fixed and stained with DAPI. Cells with chromosomes at mid-distance between the two groups of separating chromosomes were considered positive for this mitotic defect.

**Plasmids and transfections**. A list of the plasmids used in the study is provided in Supplementary Table 2. The bicistronic plasmid RFP//vimentin wt, coding for the red fluorescent protein DsRed Express2 (RFP), and untagged human vimentin wt as separate products, and the GFP fusion constructs, GFP-vimentin wt and GFP-vimentin C328S have been described in detail[11,65]. mCherry-vimentin was from Genecopoeia. The various tail truncated mutants, vimentin(1-411), (1-423), (1-448), and (1-459) were generated introducing stop codons at positions 412, 424, 449, and 460, respectively, by site-directed mutagenesis of the parent vectors using the Quikchange XL mutagenesis kit (Agilent) and the primers specified in Supplementary Table 3, following the instructions of the manufacturer. Truncated vimentin constructs showed the expected mobility in SDS-PAGE gels as well as

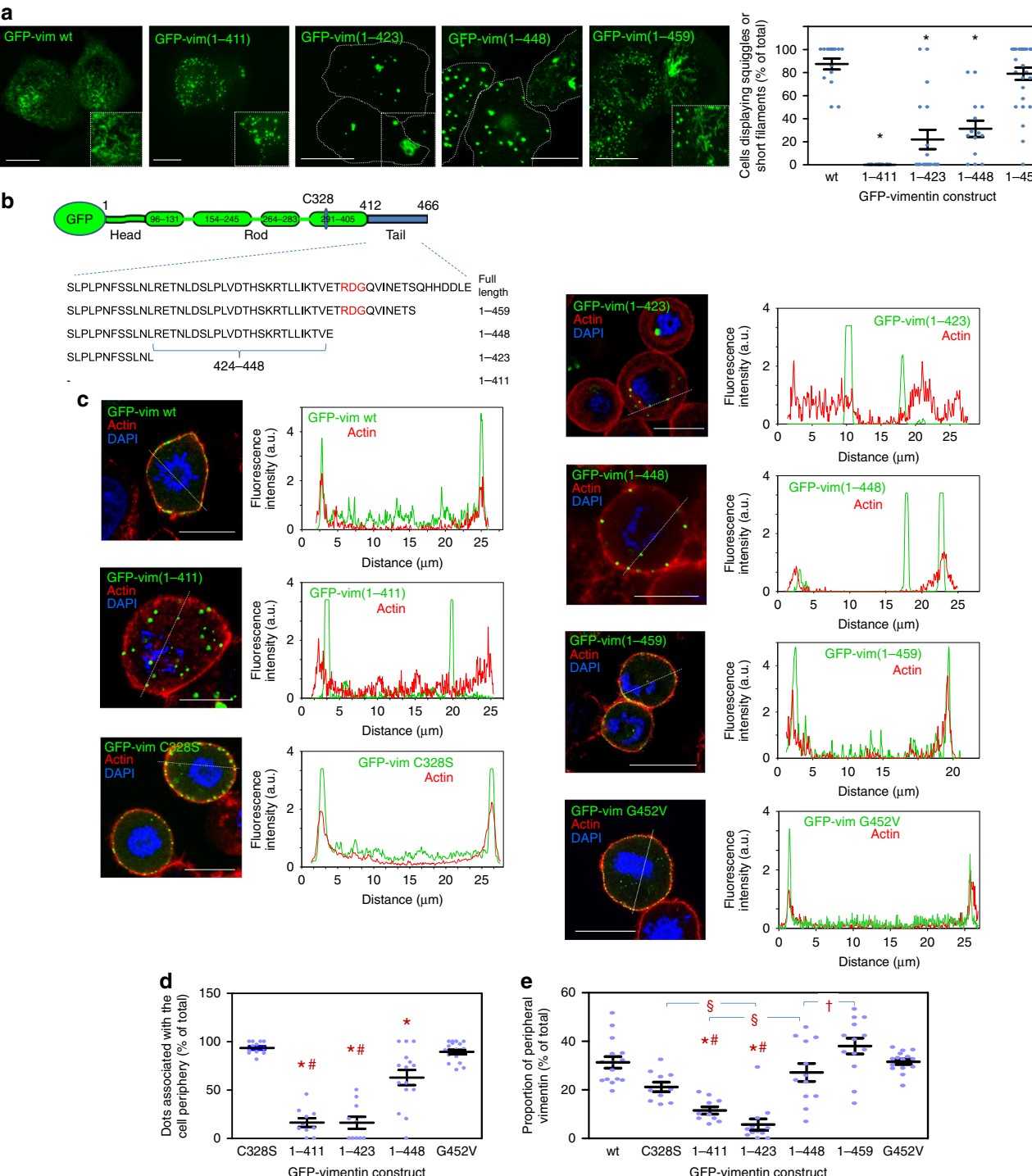

**Fig. 8** Assembly and distribution of GFP fusion constructs of truncated vimentin forms. **a** Live confocal microscopy assessment of the morphology of vimentin assemblies 48 h after transfection of SW13/cl.2 cells with the constructs schematized in panel **b**. Overall projections are shown. Insets show enlarged areas of interest. The graph (right) depicts the percentage of cells with squiggles or short filaments for every construct (*$p < 10^{-6}$ vs. wt) by two-tailed, unpaired Student's $t$-test. **c** Cells were transfected with the indicated constructs, treated overnight with 0.4 μM nocodazole in complete medium to increase the proportion of mitotic cells, fixed, and the distribution of vimentin and f-actin in mitotic cells, assessed. Nuclei were counterstained with DAPI. Single sections taken at mid-cell height are shown in all cases. For every condition, fluorescence intensity profiles for vimentin and f-actin along the dotted lines are shown on the right. **d** Proportion of vimentin dots associated with the cell periphery for every construct. In this case, GFP-vimentin C328S, which contains an intact tail domain but assembles in dots, is used as a control. GFP-vimentin(1–459) is excluded from this graph due to its organization mainly in squiggles or short filaments (*$p < 0.001$ vs. GFP-vimentin C328S and GFP-vimentin G452V; #$p < 0.001$ vs. GFP-vimentin(1–448) by ANOVA followed by Tukey's post-test for multiple comparisons). **e** Proportion of peripheral vimentin for all constructs (*$p < 0.001$ vs. GFP-vimentin wt; §$p < 0.01$; #$p < 0.01$ vs GFP-vimentin(1–448) and (1–459); †$p < 0.05$ by ANOVA followed by Tukey's post-test). The number of determinations for the experimental conditions shown in graphs from left to right was the following: **a** 15, 15, 18, 20 and 28 fields, totaling 162, 55, 77, 97 and 142 cells, respectively; **d** 12, 10, 10, 15, 19; **e** 15, 10, 10, 12, 12, 13, 20. Average values ± SEM are shown. Data are provided in the Source Data file. Scale bars, 20 μm

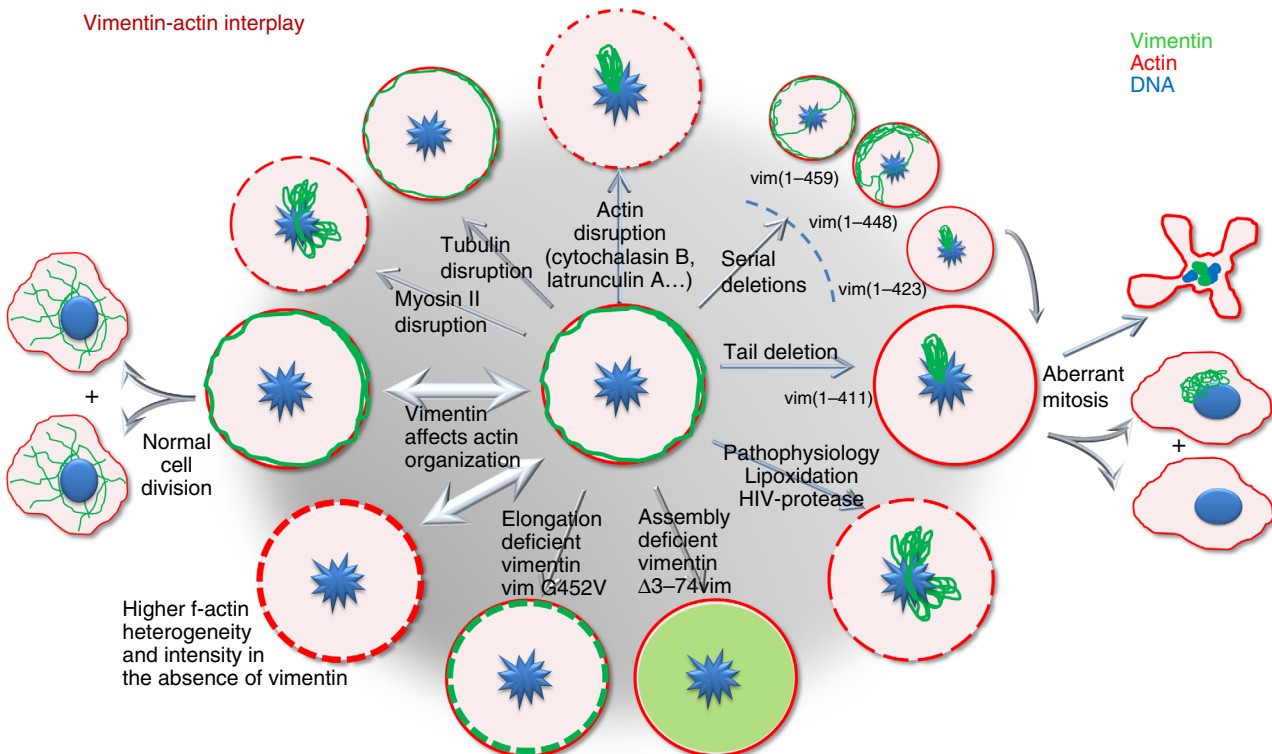

**Fig. 9** Schematic representation of the interplay between vimentin and the actin cortex in mitosis. In certain cell types in which vimentin filaments persist in mitosis (cell at the center of the figure), vimentin localizes to the actin cortex in mitosis, intertwining with cortical actin. This leads to normal mitosis, with nearly even distribution of vimentin between the two daughter cells (left). The tail segment of vimentin is required for this reorganization and normal mitosis, since vimentin distribution is progressively altered upon serial tail deletions yielding proteins spanning residues 1-459, 1-448, or 1-423, as indicated. Deletion of the whole tail segment, as in vimentin(1-411) precludes peripheral redistribution, and leads to interference with the mitotic apparatus and aberrant mitosis, with mitotic catastrophes or asymmetric partitions (right). Certain pathophysiological situations, including lipoxidation and expression of HIV-protease, mimic the distribution of tailless vimentin and impair vimentin association with the cell cortex in mitosis. A certain degree of assembly, but not the formation of full filaments, appears necessary for vimentin cortical association, since soluble vimentin constructs are diffuse and do not associate with the cell cortex, whereas elongation deficient mutants lead to peripheral particles. Comparison of vimentin-positive and vimentin-negative cells unveils that the presence of vimentin at the actin cortex in mitosis affects actin organization, influencing the heterogeneity and distribution of the f-actin signal. Conversely, disruption of actin polymerization profoundly impairs vimentin translocation to the cell periphery in mitosis (top). Disruption of myosin function also disrupts the cell cortex, whereas depolymerization of microtubules apparently has no effect. These results define an interplay of actin and vimentin in mitosis. Please note that mitotic cells are represented by circles, whereas daughter cells are shown as irregularly shaped objects. DNA is represented in blue, f-actin in red, and vimentin in green

immunoreactivity (Supplementary Fig. 5b). Thus, all constructs used were recognized by an antibody raised against full-length vimentin, which recognizes the N-terminus of the protein (anti-vim N-term), whereas an antibody against the beginning of the tail (clone V9) recognized all constructs except tailless vimentin(1-411) (Supplementary Fig. 5b, c). The GFP-vimentin(412-466) plasmid, encoding the vimentin tail domain fused to GFP, was constructed in two steps; first, an additional EcoRI site was introduced in the GFP-vimentin wt plasmid (containing the vimentin sequence cloned between the EcoRI and BamHI sites) at a position equivalent to 1692 of the vimentin mRNA sequence (accession number NM_03380.4); then, the plasmid was digested with EcoRI and re-ligated, using the Ligafast system (Promega), thus eliminating the sequence corresponding to vimentin residues 1-411. The assembly-incompetent construct RFP//vimentin Δ (3-74) was generated by introducing a SmaI site at a position equivalent to 471 of the vimentin mRNA sequence in the RFP//vimentin wt plasmid through site-directed mutagenesis. The resulting mutant plasmid was digested with SmaI and re-ligated to remove nucleotides 472 through 687, thus eliminating residues 3-74. The GFP-vimentin G452V plasmid was generated by site-directed mutagenesis using oligonucleotides specified in Supplementary Table 3 and the Nzytech mutagenesis kit. The CFP-lamin A plasmid was the gift of Dr. Vicente Andrés[66]. The vector encoding GFP-tagged HIV type I protease (pcDNA3/GFP-PR) described in ref. [67] was a gift from Nico Dantuma (Addgene plasmid #20253). Cells were transfected using Lipofectamine 2000 (Thermo Scientific). Typically, 1 μg of DNA and 3 μl of Lipofectamine 2000 were used per p35 dish. Moderate expression of vimentin in transfections using the RFP//vimentin plasmids was ensured by monitoring RFP fluorescence in microscopy experiments[11,68] and vimentin levels by immunoblot (Supplementary Fig. 5a). For overexpression of RFP//vimentin wt

or (1-411) in Vero, SW13 parental and U-251 MG astrocytoma cells, 2 μg of DNA plus 4.5 μl of Lipofectamine 2000 were used. For expression of different proportions of vimentin wt and tailless (1-411), the following plasmid amounts were used: 0.8 μg RFP//vim wt + 0.2 μg GFP-vim wt (10:0); 0.8 μg RFP//vim wt + 0.2 μg GFP-vim(1-411)(8:2); 0.4 μg RFP//vim wt + 0.4 μg RFP//vim(1-411) + 0.2 μg GFP-vim (1-411)(4:6); 0.2 μg RFP//vim wt + 0.6 μg RFP//vim(1-411) + 0.2 μg GFP-vim(1-411)(2:8). Routinely, cells were visualized 48 h after transient transfection. When indicated, cells were cultured in the presence of 500 μg/ml G-418 for generation of stably transfected cell lines. Distribution of stably transfected cell lines, as well as of engineered expression vectors, is subjected to approval of a Material Transfer Agreement by the authors' institution (CSIC, Spain).

**Fluorescence microscopy and image analysis.** Cells transfected with the various constructs were visualized live by confocal microscopy on Leica SP2 or SP5 microscopes. Images were acquired every 0.5 μm and single sections or overall projections are shown, as indicated. Scale bars are 20 μm, if not stated otherwise. For immunofluorescence, cells were fixed with 4% (w/v) paraformaldehyde for 25 min at r.t., permeabilized with 0.1% (v/v) Triton-X100 in PBS and blocked with 1% (w/v) BSA in PBS. HeLa cells were available as fixed preparations[69]. Antibodies were used at 1:200 (v/v) dilution in blocking solution. For experiments involving detection of vimentin(1-411), the monoclonal antibody recognizing the vimentin N-terminus was used for all conditions. For experiments involving selective detection of full-length vimentin or not requiring a comparison with vimentin(1-411), the V9 antibody was employed. F-actin was stained with Phalloidin-Alexa568 (Molecular Probes), following

manufacturer's instructions. Nuclei were counterstained with DAPI (3 µg/ml). Routinely, signal specificity was ensured by using controls of cells not immunostained or stained only with secondary antibodies. In addition, cells not expressing vimentin and/or constructs lacking the precise epitopes were used as controls of the specificity of the antibodies used (Supplementary Fig. 5c). Sets of samples of the experimental conditions to be compared were processed in parallel. Settings on the confocal microscope were saved and used in subsequent experiments with minor adjustments. LUT command was used to ensure non-overexposed acquisition. Direct visualization on glass-bottom culture dishes was found optimal for imaging mitotic cells, in order to preserve their spherical shape. For quantitation of impaired vimentin peripheral distribution induced by electrophilic lipids in SW13/cl.2 cells, the proportions of vimentin fluorescence present in the central area of mitotic cells (central circle comprising 60% of the total cell diameter in a single section at mid-cell height), as well as in the peripheral area, with respect to the total area were measured. For all other measurements of peripheral vimentin, a cortical ROI of ~1.5 µm thickness was manually defined with Image J (FIJI). For superresolution microscopy through stimulated emission depletion (STED), vimentin was detected with Alexa488-conjugated anti-vimentin V9 and f-actin was stained with Phalloidin–Tetramethylrhodamine B isothiocyanate from Sigma (0.25 µg/ml). Since DAPI interferes with STED, mitotic cells were spotted by observation under bright field as round cells without nuclear boundary and with the typical pattern of condensed chromosomes. Images were acquired with a confocal multispectral Leica TCS SP8 system equipped with a 3X STED module. Co-localization was analyzed with Leica and FIJI software. For quantitation of the proportion of cortical vimentin, a cortical ROI was defined by manual selection of the inner edge of the actin cortex and automatic outward extension of the selection thickness to 1 µm. In addition, the proportion of cortical vimentin colocalizing with actin was assessed within this ROI using the JacoP´s plugin and Pearson and Manders coefficients. Time-lapse microscopy was carried out in a multidimensional microscopy system Leica AF6000 LX in a humidified 5% $CO_2$ atmosphere at 37 °C. Typically, green fluorescence and DIC images were recorded. 3D-reconstructions were obtained with FIJI, Imaris or Leica software. Fluorescence intensity profiles and measurements of mean fluorescence intensity and standard deviation of pixel brightness values, to illustrate the dispersion of f-actin intensity, were obtained with FIJI. Orthogonal projections were obtained with Leica software. 2D maps from image stacks were obtained by FIJI and the free Map3-2D software developed by Sendra et al.[45] (http://www.zmbh.uniheidelberg.de//Central_Services/Imaging_Facility/Map3-2D.html), which unfolds surface information onto a single structurally connected map, using a sphere adjustment. The proportion of basal actin or vimentin with respect to the total cellular content was quantitated from image stacks acquired every 0.2 µm as the ratio of sum of the integrated intensity of the three basal planes vs. the sum of all planes, using FIJI.

**SDS–PAGE and western blot**. Cells were lysed in 20 mM Tris–HCl pH 7.5, 0.1 mM EDTA, 0.1 mM EGTA, 0.1 mM β-mercaptoethanol, containing 0.5% (w/v) SDS, 0.1 mM sodium orthovanadate and protease inhibitors (2 µg/ml each of leupeptin, aprotinin and trypsin inhibitor, and 1.3 mM Pefablock), and processed essentially as described[70]. Briefly, protein concentration in lysates was determined by the bicinchoninic acid assay. Aliquots of lysates containing the indicated amounts of total protein were denatured in Laemmli buffer for 5 min at 95 °C and separated in 10% or 15% SDS–polyacrylamide gels. Gels were transferred to Immobilon-P membranes (Millipore) using a Tris-glycine methanol three-buffer system, as recommended by the manufacturer, on a semi-dry transfer unit (Transblot, Bio-Rad). Membranes were blocked with 2% (w/v) low-fat powdered milk in T-TBS (20 mM Tris–HCl pH 7.5, 500 mM NaCl, 0.05% (v/v) Tween-20). Subsequently, membranes were incubated with primary antibodies at 1:500 (v/v) dilution and horseradish peroxidase-conjugated secondary antibodies (Dako) at 1:2000 (v/v) dilution. Proteins of interest were detected with the ECL system (GE Healthcare).

**Statistical analysis**. All experiments were repeated at least three times with similar results and measurements from distinct samples were always taken. Unless otherwise stated, all results are presented as average values ± SEM. Statistical analysis was performed with GraphPad Prism or Microsoft Excel. Statistical differences were evaluated by the unpaired two-tailed Student's $t$-test and were considered significant when $p < 0.05$, which is denoted in graphs by symbols. When indicated, ANOVA followed by Tukey post-test for multiple dataset comparisons, was used. Significance levels for every experiment are given in the figure legends. In some cases, values are grouped for conciseness. The sample size and/or the exact number of determinations for each experimental condition are given in the figure legends or in the Source Data file.

## Data availability
The data supporting all graphs presented in the study are contained in the Source Data file. Figures 1 through 8 have associated raw data. Any other data are available from the corresponding author upon reasonable request.

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

## Acknowledgements

This work has been funded by the European Union's Horizon 2020 research and innovation program under the Marie Sklodowska-Curie grant agreement number 675132, Masstrplan (http://cordis.europa.eu/project/rcn/198275_en.html), and by grants from the Spanish Ministerio de Economía y Competitividad (MINECO/FEDER, http://www.mineco.gob.es/portal/site/mineco/idi) SAF2015-68590R and RTI2018-097624-B-I00, and from Instituto de Salud Carlos III/FEDER, RETIC Aradyal RD16/0006/0021. A.V.-P. is supported by the FPI Program from MINECO, reference: BES-2016-076965. Feedback from the EU COST Action CA15214 EuroCellNet is gratefully acknowledged. We are indebted to MT Seisdedos and Dr. G. Elvira from CIB (CSIC), and Dr. S. Gutiérrez from CNB (CSIC) for help with confocal and superresolution microscopy, respectively. We thank Dr. Francisco J. Sánchez-Gómez and Prof. F.J. Cañada for helpful comments and discussion and I. Lois for help with immunofluorescence. The valuable technical assistance of M.J. Carrasco is appreciated.

## Author contributions

S.D., A.V.-P., E.N.-C., A.E.M., M.A.P., and D.P.-S. performed experiments and generated plasmids. S.D., A.V.P., D.P.-S. analyzed data and prepared illustrations. D.P.-S. and M.A.P. wrote the manuscript. D.P.-S. designed and coordinated the study.

## Additional information

**Competing interests:** The authors declare no competing interests.

