## [Peer Review File · Nature Communications]

Reviewers' comments:

Reviewer #1 (Remarks to the Author):

Duarte et al. report that overexpressed vimentin in SW13/cl.2 cells redistributes to the cell periphery during mitosis, and that this result is linked to the cortical actin cytoskeleton and appears to depend on the tail domain of vimentin. The finding is interesting and could have significant implications for how vimentin and related IFs function in cells. Consequently, this work may be potentially suitable for publication in Nature Communications.

One major issue I see with this work, however, is that the results centered on a cancer cell line, SW13/cl.2. This cell line was originally isolated as a vimentin-negative clone from the heterogeneous cancer cell line SW-13 (Ref 62). The authors here focused on expressing wide-type and modified versions of vimentin in this vimentin-negative cell line, but these cells are potentially far away from normal cell physiology, thus limiting the significance of their results.

The authors indeed also looked at several other cell types. However, results quite different from SW13/cl.2 were found –the authors themselves also noted “cortical distribution of vimentin in mitosis is cell-type dependent”. In particular, for the physiologically most relevant cell type of primary human fibroblasts, it is unclear from their single image in Fig. 4A and Supplement Fig. 3 if there is any association of the vimentin to the cell cortex at all – instead, most of the vimentin seems to be in the cytosol.

Thus, I think it would be important for the authors to show more immunofluorescence data of non-cancer cells that express endogenous vimentin, and clearly show that at least in some of them, strong association with the cell cortex is observed during mitosis. If this could be achieved, it would also be helpful to show STED data, as well as the effects of drug disruption of the actin cytoskeleton, and possibly of HIV-protease. Otherwise, the significance of their results would be limited.

Other minor issues:

1. It's unclear if “mitotic cortex” is the correct term. Should be “cell cortex during mitosis”?

2. Should add scale bars to all panels of Fig. 5 and also check several other subfigures that missed scale bars.

3. In the supplement videos, it would be helpful to add in timestamps for each frame.

Reviewer #2 (Remarks to the Author):

In this paper, Sofia et al, experts in the study of intermediate filaments, investigate the intermediate filament Vimentin network reorganization during mitosis. Using different expression constructs and different cell types, they show that in some cases Vimentin filaments redistribute close to the actin cortical network in mitosis, and that this reorganization depends on the tail domain, as tail domain deleted Vimentin protein forms clumps that fail to relocate to the actin rich cell margin during mitosis. These mutant forms then interfere with the spindle, suggesting that the cortical association of Vimentin filaments may keep them out of the way during mitosis in some systems. While the large number of observations made in the paper are interesting and novel, more work is required to determine the extent to which i) Vimentin performs a function in mitosis, ii) VimRemodelling of the Vimentin network is precisely regulated to aid mitosis and the segregation of intermediate filaments at division. Additionally, majority of the data is qualitative and needs further quantification to substantiate the papers' main conclusions.

Important issues to address:

1. For the bulk of the analysis, the authors study the role of Vimentin in a cell line which normally doesn't express the protein. The authors should explain why they chose this cell line over the others as they clearly mention that Vimentin reorganization is cell type specific. While this cell line is useful for the structure-localisation analysis, it does not help to reveal the function, timing and regulation of the dynamic remodelling of Vimentin filaments during mitosis. No information is provided about the timing or nature of Vimentin regulation in these cell lines (eg- vimentin post-translational modifications, cleavage etc).

2. Most experiments in the paper involve transient transfections. It is not clear how the authors control for the level of protein expression to compare across experiments. Also, it is not clear whether increasing level of mutant proteins themselves has an affect on protein distribution.

3. The authors need a better quantification of the mitotic defects. e.g. Duration of mitosis, cytokinesis defects, chromosome segregation defects, etc. While the authors show representative videos/images of defects in cytokinesis, chromosome segregation, these need to be quantified and correlated with respect to abnormal vimentin localization.

Specific comments:

The introduction and results sections are much too long. The authors should explain more precisely what they see as the aim and main conclusions of their study that will interest a broad audience.

Figure 1

1. Why use this cell line?
2. 1D. It would be useful to show how changes in the level of mutant proteins (in the absence of competition) affects the protein distribution.
3. Comment on 1E vs 1C. Expression of tailless vimentin affects the area/spread of vimentin network, but doesn't seem to form curly bundles in cells endogenously expressing vimentin. The phenotype is different from 1D.
4. 1G- Show a single optical section to determine whether localization is along the surface of the lobes.

Figure 2

1. According to the scheme, vimentin wt is equally distributed between the two daughter cells. However, in 1A, it seems that signal is always stronger in one of the two daughter cells in both the divisions shown. It is incorrect to claim that mutant version alone is asymmetrically segregated. Also, to make statements along these lines, the asymmetric distribution needs to be quantified.
2. Better quantification of duration of mitosis and correlation with presence of vimentin
3. Better resolution to see dynamics of vimentin reorganization during mitosis.

Figure 3

1. The representative images suggest an effect of actin perturbations on vimentin cortical localization, however, the authors need to quantify cortical/cytoplasmic vimentin in a population of cells under the different conditions to support their claim.
2. Is there more vimentin in the basal vs middle level of the cells in 3D?

Figure 4

1. 4A. The BAEC cell is in anaphase. Please ensure that the images are from comparable stages of mitosis for different cell lines as localization could change anaphase onwards.
2. Comment on the diverse localization of vimentin during mitosis in different cell lines. What does this mean for its role and regulation.
3. Desmin localization in SW13/cl.2 with and without transfection of vim-WT and vim-tailless.
4. HNE and PGA treatment results are interesting. Have these been tested in other cell lines that endogenously express vimentin?

Figure 5

1. It would help to determine how much vimentin is cortical and what % of cortical vimentin colocalizes with actin, rather than colocalization analysis for the whole cell.

Figure 6

1. Fig 5 and 6 can be combined into one figure.
2. Fig6B- Show the standard deviation of actin intensity. Vim+ cells should be compared with empty GFP control transfected cells.
3. How do you control for variability in immunostaining? Explain normalization parameters in methods.
4. 6C- quantify basal vs non-basal vimentin.
5. 6C- quantify the basal vs non-basal actin rather than only basal to take variability in immunostaining into account.
6. Increase the number of cells to obtain robust conclusions.
7. Vim411 orthogonal sections: Comment on the prominent ring of actin inside the cells expressing this mutant version which is absent in vim-WT expressing cells or non-transfected cells. Does Vim411 affect non-cortical actin localization in the cells?

Figure 7

1. 7B- RFP appears to be enriched in the nucleus?
2. Quantify, vimentin area vs cell area.
3. Quantify mitosis defect correlation with presence of vimentin near chromosome.

Figure 8

1. Quantify cortical vs non-cortical vim for different constructs.
2. 8C - corresponds to mitotic or non-mitotic cells? Its not clear from the text.

Reviewer #3 (Remarks to the Author):

The role of vimentin in cell division is an important and grossly understudied area of cell biology. Although there have been a few papers related to this topic over the last several decades, little has been accomplished in recent years. The paper by Duarte et al titled “Vimentin filaments interact with the mitotic cortex allowing normal cell division””suggests that vimentin filaments redistribute to the cell periphery during mitosis, forming a robust scaffold interwoven with cortical actin and affecting the mitotic cortex properties.” They further claim that it is the tail domain of vimentin which is responsible for the redistribution to the cell cortex.

Critique.

The results section contains evidence for the role of vimentin in mitotic cells based exclusively on the expression of either GFP/RFP tagged wild type vimentin or similarly tagged tail-less vimentin (their 1-411 construct) in cells that do not express endogenous vimentin. The majority of their experiments utilize SW13/cl.2 adrenal carcinoma cells and

MCF7 cells derived from mammary gland tumors which retain their epithelial cell characteristics. It appears that all of their conclusions are based upon cells ectopically expressing vimentin, mainly in its truncated tail-less form. A detailed analysis of a cell type which expresses a normal network of vimentin intermediate filaments, and no other cytoskeletal members of the IF protein family, would represent an important control for all of the experimental data presented. Furthermore, although it has been accepted that the SW13 cells do not express endogenous vimentin and likely no other cytoskeletal intermediate filaments, it is very well known that MCF 7 cells contain extensive arrays of keratin containing intermediate filaments. However, there is no mention of this in the results. Nor is there an analysis of the impact of overexpressing various vimentin constructs on the endogenous keratin network which has been shown to interact with cortical actin in cells similar to MCF7.

The results section emphasizes mainly the role of the abnormal construct, vimentin 1-411 which has been shown by others to permit the assembly of vimentin into structures with the overall features of 10 nm intermediate filaments. However, there have been no exhaustive studies showing that this construct behaves like full length wild type vimentin in null backgrounds. This is important as the authors claim that they have achieved numerous important insights into the distribution and function of vimentin based almost exclusively on their microscopic studies of cells in which the only form of vimentin expressed is the 1-411 truncated protein. There is no consideration of the potential deleterious effects of the overexpression of the various truncated vimentins which include 1-411, 1-423, 1-448, 1-459, etc. The expression levels should be determined by quantitative immunoblotting, but the only blots shown are those in the supplement which demonstrate that their various anti-vimentin antibodies react with one or more of the truncated mutants. This is very important. For example, the head domain of vimentin which is intact in the 1-411 construct contains many phosphorylation sites (in human over 40), and many of these sites are phosphorylated by major kinases involved in signal transduction (e.g., PKA, PKC, ROK). Therefore over expression of these domains could act as a competitive inhibitor of numerous signal transduction pathways. In addition, although a complete deletion of the tail domain of vimentin may permit assembly of filaments, there is evidence that small deletions or point mutations in tail domains, such as the RDG sequence, actually cause defects in assembly of intermediate filaments. So justification for the use of the 1-423, 1-448, 1-459 must be rigorously tested using in vitro assembly assays to determine their impact on the assembly of vimentin filaments.

Overall the conclusions are drawn from images of cells expressing tagged truncated vimentins

and in some cases WT GFP/RFP tagged vimentin. However, many of these are of poor quality. One of numerous examples is Figure 4D.

The authors also fail to acknowledge that it is well known that drugs such as nocodazole alter the distribution of many types of IF including vimentin. This causes the aggregation of

vimentin into large bundles and coils--so what are the side effects of long term treatments used to increase mitotic cells? Actually there is no convincing evidence presented that the cells depicted in Fig 5 are mitotically arrested.

The authors employ the use of many drugs such as nocodazole, latrunculin A, C3 toxin, jaspalkinolide, blebbistatin, etc. The impact of these drugs is difficult to evaluate as there appear to be no statistics; just line scans displaying fluorescence intensity. How many cells were assayed? How was the data analyzed statistically? These issues must be addressed.

Vimentin filaments interact with the actin cortex in mitosis allowing normal cell division

Sofia Duarte, Álvaro Viedma-Poyatos, Elena Navarro-Carrasco, Alma E. Martínez, María A. Pajares, Dolores Pérez-Sala*

RESPONSES TO REVIEWERS' COMMENTS:

We sincerely appreciate the comments of the reviewers, which have greatly contributed to strengthen and clarify the observations in the paper.

Reviewer #1 (Remarks to the Author):

Duarte et al. report that overexpressed vimentin in SW13/cl.2 cells redistributes to the cell periphery during mitosis, and that this result is linked to the cortical actin cytoskeleton and appears to depend on the tail domain of vimentin. The finding is interesting and could have significant implications for how vimentin and related IFs function in cells. Consequently, this work may be potentially suitable for publication in Nature Communications.

One major issue I see with this work, however, is that the results centered on a cancer cellline, SW13/cl.2. This cell line was originally isolated as a vimentin-negative clone from the heterogeneous cancer cell line SW-13 (Ref 62). The authors here focused on expressing wide-type and modified versions of vimentin in this vimentin-negative cell line, but these cells are potentially far away from normal cell physiology, thus limiting the significance of their results. We thank the reviewer for the constructive summary of our work. Following the reviewers' advice, the revised version of the manuscript now includes additional information obtained from various cell types that ensures the significance of our work.

The authors indeed also looked at several other cell types. However, results quite different from SW13/cl.2 were found –the authors themselves also noted “cortical distribution of vimentin in mitosis is cell-type dependent”. In particular, for the physiologically most relevant cell type of primary human fibroblasts, it is unclear from their single image in Fig. 4A and Supplement Fig. 3 if there is any association of the vimentin to the cell cortex at all –instead, most of the vimentin seems to be in the cytosol.

Thus, I think it would be important for the authors to show more immunofluorescence data of non-cancer cells that express endogenous vimentin, and clearly show that at least in some of

them, strong association with the cell cortex is observed during mitosis. If this could be achieved, it would also be helpful to show STED data, as well as the effects of drug disruption of the actin to skeleton, and possibly of HIV-protease. Otherwise, the significance of their results would be limited.

We thank the reviewer for this important comment. In the revised version of the manuscript we have included additional non-cancer cell types expressing endogenous vimentin, which now comprise bovine aortic endothelial cells (BAEC), human dermal fibroblasts, human renal

mesangial cells (NHMC)(Fig. 4a) and fibroblast-like Vero cells (Fig. 1e, Fig. 3h, etc.). For easier reference, we have included a table specifying the cell types used in the study and their characteristics, as Supplementary Table I (please, see below).

In particular, in Vero cells, we have confirmed the strong association of endogenous vimentin with the cell cortex using STED(Fig. 5b). Moreover, we have confirmed the effect of actin disruption (Fig. 3h) and of lipoxidation (Fig. 4d). Additionally, we have explored the effect of expressing HIVprotease on endogenous vimentin distribution, and we observed strong condensation or loss of immunoreactivity in interphase cells (Fig. 4g), although in this case, no mitotic cells expressing the protease could be found.

Nevertheless, we would like to emphasize that the study of vimentin distribution in cancer cell lines is also significant since vimentin is a marker and key player of epithelial mesenchymal transition. In this regard, we have also confirmed that vimentin is associated with the actin cortex in the widely used cancer cell model HeLa. These results have also been incorporated into the manuscript (Fig. 4a and Supplementary Fig. 2).

Other minor issues:

1. It's unclear if "mitoticcortex" is the correct term. Should be "cell cortex during mitosis"? We thank the reviewer for this comment. Although the term "mitotic cortex" has been used in previous publications, we agree that mitotic cell cortex or cell cortexduring mitosis would be more appropriate. Therefore, we have corrected this term in the revised version of the manuscript.

2. Should add scale bars to all panels of Fig. 5 and also check several other subfigures that missed scale bars.

As requested by the reviewer we have added scale bars to Fig. 5 and other subfigures. Nevertheless, we have not added them in insets showing enlarged areas of images if they were already present in the primary images.

3. In the supplement videos, it would be helpful to add in time stamps for each frame. This has been added.

Reviewer #2 (Remarks to the Author):

In this paper, Sofia et al, experts in the study of intermediate filaments, investigate the intermediate filament Vimentin network reorganization during mitosis. Using different expression constructs and different cell types, they show that in some cases Vimentin filaments redistribute close to the actin cortical network in mitosis, and that this reorganization depends on the tail domain, as tail domain deleted Vimentin protein forms clumps that fail to relocate to the actin rich cell margin during mitosis. These mutant forms then interfere with the spindle, suggesting that the cortical association of Vimentin filaments may keep them out of the way during mitosis in some systems. While the large number of observations made in the paper are interesting and novel, more work is required to determine the extent to which i) Vimentin performs a function in mitosis, ii) Vimentin Remodelling of the Vimentin network is precisely regulated to aid mitosis and the segregation of intermediate filaments at division. Additionally, majority of the data is qualitative and needs further quantification to substantiate the papers' main conclusions.

We thank the reviewer for this insightful summary of our work. In the revised version of the manuscript, and thanks to the suggestions of the reviewers, we provide additional information strengthening the concept that vimentin performs a function in mitosis. As the reviewer summarizes, we propose that vimentin reorganization plays, at least, a permissive role in mitosis by either undergoing disassembly, or associating with the cell cortex to avoid interference with the mitotic apparatus. We show that in several cell types, including primary endothelial cells, fibroblasts, Vero cells, HeLa, human mesangial cells, and astrocytoma cells, vimentin filaments do not completely disassemble in mitosis. In these cell types, vimentin shows a variable degree of association/anchorage with the cell cortex in mitotic cells. Moreover, the presence of vimentin at the cell cortex in mitosis influences actin distribution. This is even clearer in the revised version of the manuscript thanks to the inclusion of new analysis requested by this reviewer, which provide a relative assessment of actin distribution within each cell.

The mechanisms regulating vimentin remodeling, and particularly disassembly, in mitosis, involve phosphorylation and protein-protein interactions. These processes have been previously addressed in several works (refs 17, 42, 43 and 53 in the manuscript). Indeed, a vimentin mutant, bearing mutations at several N-terminal residues that get phosphorylated in mitosis, induces cytokinetic defects (ref 43). In light of these previous evidences we have not focused on these aspects. Nevertheless, they have been considered in more detail in the text of the revised manuscript (please, see below). Finally, additional quantitative data have been included in the manuscript that further strengthen the conclusions.

Important issues to address:

1. For the bulk of the analysis, the authors study the role of Vimentin in a cell line which normally doesn't express the protein. The authors should explain why they chose this cell line over the others as they clearly mention that Vimentin reorganization is cell type specific. While this cell line is useful for the structure-localisation analysis, it does not help to reveal the function, timing and regulation of the dynamic remodelling of Vimentin filaments during

mitosis. No information is provided about the timing or nature of Vimentin regulation in these cell lines (eg- vimentin post-translational modifications, cleavage etc).

We thank the reviewer for these comments. The reason to use the SW13/cl.2 cell line is its lack of cytoplasmic intermediate filaments, which allows dissecting the effect of vimentin mutations without the interference of endogenous vimentin or other intermediate filament proteins. This aspect is now made clearer in the revised version of the manuscript (section 1 of the Results). Nevertheless, throughout the manuscript, we have used different cell types, including MCF7 breast carcinoma cells, U-251 MG astrocytoma cells, bovine aortic endothelial cells, HeLa, and Vero monkey fibroblasts in which the main findings of the paper are confirmed, including persistence of vimentin filaments in mitosis with significant interaction at the actin cortex. A supplementary table detailing the cell types used has been included in the revised version of the manuscript (Supplementary Table I). Moreover, thanks to the reviewer's suggestions, the impact of vimentin in mitosis is now better characterized (please see below for details).

The timing and regulation of vimentin in mitosis has been previously studied in several cell lines (refs. 17, 42, 43 and 53). Most of the available information refers to its phosphorylation-induced disassembly, which is attributed to phosphorylation of residues present in the amino terminus of the protein. In addition, the presence of proteins capable of heteropolymerize with vimentin, such as nestin, has been involved in vimentin disassembly or filament persistence. Thus, persistence of vimentin filaments in mitosis is attributed to lack of nestin or of phosphorylation. In this work, we have not addressed the mechanisms involved in vimentin solubilization or filament persistence but followed the distribution of these filaments in mitosis, focusing on the cell types in which disassembly does not take place.

We agree with the reviewers on the importance of vimentin posttranslational modifications (PTM) for the regulation of this protein.

[Redacted]

Regarding vimentin cleavage, we have indeed attempted to characterize vimentin products in several cell lines. However, this has been precluded by the low proportion of mitotic cells among those transfected with HIV protease. In fact, in Vero cells, it was not possible to detect mitotic cells transfected with the protease.

2. Most experiments in the paper involve transient transfections. It is not clear how the authors control for the level of protein expression to compare across experiments. Also, it is not clear whether increasing level of mutant proteins themselves has an affect on protein distribution.

The use of the bicistronic plasmids RFP//vimentin allows assessing the level of RFP by monitoring the intensity of red fluorescence, which correlates with the level of vimentin expressed. Additionally, we have confirmed that cells stably transfected with these plasmids, express vimentin at levels well below those detected in primary fibroblasts (as illustrated in Supplementary Fig. 5a), for which we are confident that the effects observed are not the result of abnormal protein distribution due to overexpression. In addition, as requested by the reviewer and detailed below (response to point 2 of Fig. 1), we have confirmed that expression of different levels of mutant vimentin yields similar patterns.

3. The authors need a better quantification of the mitotic defects. e.g. Duration of mitosis,

cytokinesis defects, chromosome segregation defects, etc. While the authors show representative videos/images of defects in cytokinesis, chromosomes egregation, these need to be quantified and correlated with respect to abnormal vimentin localization.

Thanks to the requests of the reviewer, in the revised version of the manuscript we have significantly improved the quantitative assessment of the mitotic defects, including duration of mitosis, chromosomes egregation defects and appearance of cells with multiple aberrant nuclei (Fig. 2 and 7). The results obtained strengthen our previous conclusions as well as the correlation with respect to abnormal vimentin localization.

Specific comments:

The introduction and results sections are much too long. The authors should explain more precisely what they see as the aim and main conclusions of their study that will interest a broad audience.

The introduction has been shortened and the focus improved. The results have been exposed more concisely. Nevertheless, the extension of this section has not been reduced due to the inclusion of substantial new information.

Figure 1

1. Why use this cell line?

This cell line is a well characterized model which does not express any cytoplasmic intermediate filament, thus allowing the study of single intermediate filament proteins or their combinations, either in their wildtype or mutant forms. Indeed, it has been widely used in the field of intermediate filament research, with at least 52 references appearing in PubMed with the terms "SW13 intermediate filaments or vimentin". The reasons for using this cell line have been clarified in the manuscript (Section 1 of the Results).

2. 1D. It would be useful to show how changes in the level of mutant proteins (in the absence of competition) affects the protein distribution.

The experiment suggested by the reviewer has been performed and the results are shown in Fig. 1d, upper panels. As it can be observed, the curly pattern of vimentin(1-411) does not appreciably change with the amount of construct transfected.

3. Comment on 1E vs 1C. Expression of tailless vimentin affects the area/spread of vimentin network, but doesn't seem to form curly bundles in cells endogenously expressing vimentin. The phenotype is different from 1D.

We thank the reviewer for this comment. In this regard, we would like to clarify that what we explore in Fig. 1e is the organization of endogenous full-length vimentin in the presence of overexpressed tailless vimentin. The immunofluorescence is performed with an antibody against the C-terminal segment of vimentin for which tailless vimentin is not detected in these images. However, as noted by the reviewer, the disruption of endogenous vimentin adopts slightly different patterns in the various experimental models. Indeed, in U-251 MG astrocytoma cells endogenous vimentin collapses into a perinuclear ring in most cells, whereas in Vero and in parental SW13 cells, both, accumulations of thick bundles and perinuclear bundles can be observed. Potential reasons for these differences could rely in the different

composition of the cytoplasmic intermediate filament network in the various cell types. In SW13/cl.2 cells, only transfected vimentin wt or (1-411) are present. Parental SW13 cells only express vimentin. Vero cells express at least vimentin and keratins. U-251 MG astrocytoma express various cytoplasmic intermediate filaments, including vimentin, GFAP, nestin, synemin and cytokeratins in variable proportions (Leduc (2017) J Cell Biol 216,1689; Zhou (2000) Exp Cell Res 254, 269; Pan (2008) FASEB J 22, 3196; Choi(2018) Mol Cell Proteomics 17, 1948). Notably, GFAP, nestin and synemin can copolymerize with vimentin. Therefore, the organization of vimentin(1-411) or the disruption of endogenous vimentin do not necessarily follow the same pattern in every cell type.

4. 1G- Show a single optical section to determine whether localization is along the surface of the lobes.

Images shown in Fig. 1g are indeed single optical sections (whereas overall projections are shown in insets in Fig 1g). In these images it is clear that vimentin bundles localize between the lobes or in grooves of the nuclear membrane. We apologize if this was not clear enough in the previous version of the manuscript. This has been clarified in the Figure legend of the revised manuscript. Moreover, an additional image has been included in Fig. 1g to better illustrate this point.

Figure 2

1. According to the scheme, vimentin wt is equally distributed between the two daughter cells. However, in 1A, it seems that signal is always stronger in one of the two daughter cells in both the divisions shown. Its incorrect to claim that mutant version alone is asymmetrically segregated. Also, to make statements along these lines, the asymmetric distribution needs to be quantified.

We thank the reviewer for this observation. The scheme shown in the previous version of the manuscript was indeed a simplified view of the process. In the revised version, the scheme has been modified to depict daughter cells with a slightly different content of vimentin. In addition, the content of vimentin in the two daughter cells for every construct has been quantitated and a graph is now presented as Fig. 2d, confirming that, indeed, the distribution of vimentin between daughter cells is much more dissimilar in the case of the mutant.

2. Better quantification of duration of mitosis and correlation with presence of vimentin

As requested by the reviewer, the duration of mitosis according to the presence/absence of vimentin wt or (1-411) has been quantitated and the results are shown in Fig. 2c.

3. Better resolution to see dynamics of vimentin reorganization during mitosis.

In order to provide better resolution of vimentin distribution we now show images from the green channel (vimentin fluorescence) in gray scale.

In addition, as suggested by the reviewer in general comment 3, chromosome segregation defects have been assessed by estimating the proportion of cells showing lagging chromosomes in anaphase and the results are shown in Fig. 2f.

Figure 3

1. The representative images suggest an effect of actin perturbations on vimentin cortical localization, however, the authors need to quantify cortical/cytoplasmic vimentin in a population of cells under the different conditions to support their claim.

We thank the reviewer for this suggestion. We have quantitated the proportion of cortical/cytoplasmic vimentin upon treatment with the various disrupting agents and the results are shown in Fig. 3g.

2. Is there more vimentin in the basal vs middle level of the cells in 3D?

To answer this question, we have reviewed these images and confirmed that in cells treated with actin disrupting agents there is no more vimentin at the basal level than at the middle level.

Figure 4

1. 4A. The BAEC cellisina phase. Please ensure that the images are from comparable stages of mitosis for different cell lines as localization could change anaphase onwards.

We thank the reviewer for this observation. We have changed the image of BAEC. In addition, following the suggestions of the reviewers, the panel of primary cells shown in this Figure has been expanded and now includes BAEC, primary human fibroblasts and human mesangial cells. These results are provided in Fig 4a.

2. Comment on the diverse localization of vimentin during mitosis in different cell lines. What does this mean for its role and regulation.

Although a brief comment on the cell-type dependence of vimentin distribution in mitosis was present in the previous version of the manuscript, following the request of the reviewer, we have discussed this point in more detail in page16. Briefly, the main difference between cell types noted to date is related to vimentin disassembly or persistence to various extents. Vimentin disassembly has been attributed to the cooperation between phosphorylation and interaction with nestin.

3. Desmin localization in SW13/cl.2 with and without transfection of vim-WT and vim-tailless. We have performed the experiment requested by the reviewer.

[Redacted]

4. HNE and PGA treatment results are interesting. Have these been in tested in other cell lines that endogenously express vimentin?

We have previously assessed the effects of PGA and HNE in astrocytoma cells (Viedma-Poyatos et al., 2018, Free Rad Biol Med 120:380) and renal mesangial cells (Pérez-Sala et al., 2015, Nat Commun.6,7287).In both cases we evidenced the condensation of vimentin filaments near the center of the cell. Nevertheless, we had not assessed their distribution in mitosis. In the revised version of the manuscript, following the inquiry of the reviewer we have addressed this point in Vero cells, which endogenously express vimentin. Treatment of these cells with lipoxidative agents results in a decrease in the proportion of vimentin localized at the cell periphery in mitosis, together with an increase in bundling near the dividing chromosomes (shown in Fig. 4d), similar to that elicited in SW13/cl.2 cells transfected with vimentin wt (Fig. 4c).

Figure 5

1. It would help to determine how much vimentin is cortical and what % of cortical vimentin colocalizes with actin, rather than colocalization analysis for the whole cell. Following the advice of their viewer we have calculated the proportion of both cortical-actin and cortical vimentin, and the results are shown in Fig. 5d. Moreover, the specific colocalization of cortical vimentin with actin is given as overlap and Pearson coefficients (Fig. 5e).

Figure 6

1. Fig 5 and 6 can be combined into one figure.

We would agree with the reviewer about this possibility for the former version of the manuscript. Nevertheless, new information has been included in both figures in the present version. Therefore, we consider that a clear presentation of the results requires two separate figures.

2. Fig6B- Show the standard deviation of actin intensity. Vim+ cells should be compared with empty GFP control transfected cells. We thank the reviewer for this suggestion. As in this experiment vimentin was cloned in the bicistronic pIRES-DsRed-Express2vector (RFP// vector), in the revised version of the manuscript we have included the values obtained from cells transfected with the “empty” RPF//vector.

3. How do you control for variability in immunostaining? Explain normalization parameters in methods.

For immunofluorescence, controls of cells not immunostained or stained only with secondary antibodies were always used when setting-up the procedure in order to ensure specificity of the signals. In addition, cells not expressing vimentin and/or constructs lacking the precise epitopes were used as controls of the specificity of the antibodies used, as shown in Supplementary Fig. 5c. For f-actin staining, Phalloidin-TRITC was used always at the same dilution using aliquots of a stock kept at -20°C. Sets of samples of the experimental conditions to be compared were processed in parallel. Settings on the confocal microscope were saved and used in subsequent experiments with minor adjustments. LUT command was used to ensure non-overexposed acquisition. A summary of these procedures has been included in Methods. In the manuscript, f-actin intensity measurements appear mainly Fig. 6e. These results are now additionally supported by the determinations of relative basal f-actin abundance suggested by the reviewer (Fig. 6d).

4. 6C- quantify basal vs non-basal vimentin.

This quantitation has been performed and is depicted in panel 6d, showing that basal vimentin is clearly more abundant in cells expressing vimentin wt.

5. 6C- quantify the basal vs non-basal actin rather than only basal to take variability in immunostaining into account.

This feature has been quantitated and is depicted in Fig. 6d, showing that the proportion of basal vs total actin is lower in cells expressing vimentin wt.

6. Increase the number of cells to obtain robust conclusions.

More observations are now included in graphs 6b, 6d and 6e.

7. Vim411 orthogonal sections: Comment on the prominent ring of actin inside the cells expressing this mutant version which is absent in vim-WT expressing cells or non-transfected cells. Does Vim411 affect non-cortical actin localization in the cells?

The prominent actin ring inside cells is indeed an intriguing observation. This internal actin ring has been previously observed in HeLa cells (Lu (2014) PLoS ONE 9:e102547), and has been hypothesized to play a role in spindle positioning. This ring appears to form at a position close to the spindle, and it is connected to cortical actin by a mesh of thin cytoplasmic actin filaments (Lu et al., 2014). Under our conditions we observe a prominent ring above all in cells transfected with vimentin(1-411) (Fig. 6c) and vimentin(1-423) (Fig. 7e). However, it is also visible in cells not expressing vimentin(1-411) (Fig. 6c), and hence, it does not seem to appear as a consequence of vim(1-411) expression. Moreover, the intensity of this structure is not homogeneous in the cell population. Therefore, defining its nature and the relationship to vimentin, if any, will be a pertinent and highly interesting topic for future studies.

Figure 7

1. 7B- RFP appears to be enriched in the nucleus?

RFP is a small protein that distributes both in the nucleus and the cytoplasm, although the proportion observed in both compartments varies from cell to cell even in the same experiment. Thus, we did not observe a consistent nuclear enrichment of the RFP transfection reporter protein.

2. Quantify, vimentin area vs cell area.

The vimentin area vs the cell area has been quantitated in interphase SW13/cl.2 cells for the constructs shown and is depicted in panel 7n.

3. Quantify mitosis defect correlation with presence of vimentin near chromosome.

These features have been quantitated and the results, depicted in Fig. 7o, show that the average percentage of cells with vimentin near chromosomes for every construct correlates with the mitotic defects as reflected by the longer duration of mitosis and the presence of cells with multiple or aberrant nuclei (Fig. 7o).

Figure 8

1. Quantify cortical vs non-cortical vim for different constructs.

This has been quantitated and is shown in Fig 8e.

2. 8C - corresponds to mitotic or non-mitotic cells? Its not clear from the text.

Figure 8 has been simplified for clarity by removing the interphase cells from panel c. Thus, panels in 8c now correspond only to mitotic cells.

Reviewer #3 (Remarks to the Author):

The role of vimentin in cell division is an important and grossly understudied area of cell biology. Although there have been a few papers related to this topic over the last several decades, little has been accomplished in recent years. The paper by Duarte et al titled “Vimentin filaments interact with the mitotic cortex allowing normal cell division”” suggests that vimentin filaments redistribute to the cell periphery during mitosis, forming a robust scaffold interwoven with cortical actin and affecting the mitotic cortex properties.” They further claim that it is the tail domain of vimentin which is responsible for the redistribution to the cell cortex.

We thank the reviewer for raising the importance of this topic and for the summary of our work. We also acknowledge all the detailed and insightful comments that have been addressed or clarified as detailed below.

Critique.

The results section contains evidence for the role of vimentin in mitotic cells based exclusively on the expression of either GFP/RFP tagged wild type vimentin or similarly tagged tail-less vimentin (their 1-411 construct) in cells that do not express endogenous vimentin.

We apologize if the description of the constructs used was not clear enough in the previous version of the manuscript. Nevertheless, we would like to emphasize that the majority of the experiments have been carried out with bicistronic constructs that express untagged forms of the proteins and RFP as separate transcripts. RFP, then, is useful as a control of transfection and as an index of the expression levels (please, see Scheme in Fig. 1b). To further clarify this point, we now list all the constructs used in Supplementary table II (appended below for the reviewer’s convenience).

Supplementary Table II. Constructs used in the study

Name	Abbreviation	Nature	Products expressed	Tag
1 pIRES-DsRed Express2	RFP//	Bicistronic	RFP	NA
2 pIRES-DsRed Express2-vimentin wt	RFP//vim wt	Bicistronic	RFP and vimentin aa 1 to 466 (separate products)	None
3 pIRES-DsRed Express2-vimentin(1-411)	RFP//vim(1-411)	Bicistronic	RFP and vimentin aa 1 to 411 (separate products)	None
4 pIRES-DsRed Express2-vimentin(1-423)	RFP//vim(1-423)	Bicistronic	RFP and vimentin aa 1 to 423 (separate products)	None
5 pIRES-DsRed Express2-vimentin(1-448)	RFP//vim(1-448)	Bicistronic	RFP and vimentin aa 1 to 448 (separate products)	None
6 pIRES-DsRed Express2-vimentin(1-459)	RFP//vim(1-459)	Bicistronic	RFP and vimentin aa 1 to 459 (separate products)	None
7 pIRES-DsRed Express2-vimentin Δ3-74	RFP//vimΔ3-74	Bicistronic	RFP and vimentin aa (1-2)/(75-466) (separate products)	None
8 pEGFP-C1	GFP	Vector	GFP	NA
9 pEGFP-C1-vimentin wt	GFP-vim wt	Fusion	GFP fused to vimentin wt	GFP
10 pEGFP-C1-vimentin(1-411)	GFP-vim(1-411)	Fusion	GFP fused to aa 1 to 411 of vimentin	GFP
11 pEGFP-C1-vimentin(1-423)	GFP-vim(1-423)	Fusion	GFP fused to aa 1 to 423 of vimentin	GFP
12 pEGFP-C1-vimentin(1-448)	GFP-vim(1-448)	Fusion	GFP fused to aa 1 to 448 of vimentin	GFP
13 pEGFP-C1-vimentin(1-459)	GFP-vim(1-459)	Fusion	GFP fused to aa 1 to 459 of vimentin	GFP
14 pEGFP-C1-vimentin C328S	GFP-vim C328S	Fusion	GFP fused to vimentin C328S	GFP
15 pEGFP-C1-vimentin G452V	GFP-vim G452V	Fusion	GFP fused to vimentin G452V	GFP
16 pEGFP-C1-vimentin(412-466)	GFP-vim(412-466)	Fusion	GFP fused to aa 412 to 466 of vimentin	GFP
17 mCherry-vimentin wt	mCherry-vimentin wt	Fusion	mCherry-vimentin wt	mCherry
18 pECFP-C1-Lamin A	CFP-Lamin A	Fusion	CFP-Lamin A	CFP
19 pDNA3/GFP-PR	GFP-PR	Fusion	GFP-HIV type I protease	GFP

The majority of their experiments utilize SW13/cl.2 adrenal carcinoma cells and MCF7 cells derived from mammary gland tumors which retain their epithelial cell characteristics. It appears that all of their conclusions are based upon cells ectopically expressing vimentin, mainly in its truncated tail-less form.

We thank the reviewer for this comment. To address the reviewers' concern, in the current version of the manuscript we have used additional cell types. For easier reference, we have included a table (Supplementary Table I), listing the cell types used in the study, which now include primary cells, non-tumoral cell lines and tumoral cell lines, of which, some express and some do not express vimentin. Please refer to this table for detailed information. Moreover, we have reproduced the key observations of the manuscript in cells expressing endogenous vimentin, including Vero cells and parental SW13 cells. Regarding the constructs studied, we would like to summarize their use in the manuscript as follows:

Fig. 1 and 2 report the distribution of both vimentin full-length and truncated vimentin.

Fig. 3 contains information about the two constructs, but mainly about full-length vimentin.

Fig 4 and 5 are devoted to vimentin wt.

Figure 6 contains information about the two constructs.

Figs 7 and 8 compare the various truncated constructs, as well as the GFP-fusion constructs of vimentin wt and point mutants C328S and G452V (Fig. 8).

A detailed analysis of a cell type which expresses a normal network of vimentin intermediate filaments, and no other cytoskeletal members of the IF protein family, would represent an important control for all of the experimental data presented.

We thank the reviewer for this suggestion. Regarding IF proteins expression, as far as we know, all cells express nuclear lamins and most cells express several cytoplasmic IF. To the best of our knowledge, and according to Sarria et al., 1990, 1992, the only cell line expressing vimentin as the only cytoplasmic IF, are parental SW13 cells, a high proportion of which contain endogenous vimentin. Therefore, these cells could be considered an appropriate control. Thus, among the cells used in the revised version, we present results from parental SW13 cells expressing endogenous vimentin, where we have confirmed: i) the peripheral distribution of vimentin in mitosis; ii) its susceptibility to actin disruption (Supplementary Fig. 1d); and iii) the disrupting effect of tailless vimentin expression on endogenous vimentin distribution (Fig. 1e). The results obtained confirm those observed through the expression of vimentin in the vimentin-deficient SW13/cl.2 cells.

Furthermore, although it has been accepted that the SW13 cells do not express endogenous vimentin and likely no other cytoskeletal intermediate filaments, it is very well known that MCF 7 cells contain extensive arrays of keratin containing intermediate filaments. However, there is no mention of this in the results.

We thank the reviewer for this observation. In the revised version of the manuscript we have improved the information on other cytoplasmic IF present in the cell types used, namely, GFAP and nestin in astrocytoma cells and keratins in MCF7 cells (section 1 of the Results).

Nor is there an analysis of the impact of overexpressing various vimentin constructs on the endogenous keratin network which has been shown to interact with cortical actin in cells similar to MCF7.

We thank the reviewer for this comment. Indeed, a subplasmalemmal rim of keratin has been defined in epithelial cells that attaches at desmosomes (Quinlan et al., 2017, J Cell Sci, 130, 3437), and keratin precursors have been reported to appear close to focal adhesions (Windoffer et al., 2006, J Cell Biol 173, 341). ...

[Redacted]

The results section emphasizes mainly the role of the abnormal construct, vimentin1-411 which has been shown by others to permit the assembly of vimentin into structures with the overall features of 10 nm intermediate filaments. However, there have been no exhaustive studies showing that this construct behaves like full length wild type vimentin in null backgrounds. This is important as the authors claim that they have achieved numerous important insights into the distribution and function of vimentin based almost exclusively on their microscopic studies of cells in which the only form of vimentin expressed is the 1-411 truncated protein.

As detailed above, the results section covers both observations on vimentin wt distribution and on the truncated form(s). Indeed, all figures, except Fig. 7, present results obtained with full length vimentin. Therefore, our main interest is not the “role” of the truncated forms themselves, but their use as tools to elucidate the importance of the tail domain in the association of vimentin with the cortex and the consequences of the disruption of this association. In order to obtain insight into the distribution and function of vimentin, we have also used vimentin-expressing and vimentin-null cells.

There is no consideration of the potential deleterious effects of the overexpression of the various truncated vimentins which include 1-411, 1-423, 1-448, 1-459, etc. The expression levels should be determined by quantitative immunoblotting, but the only blots shown are those in the supplement which demonstrate that their various anti-vimentin antibodies react with one or more of the truncated mutants.

We thank the reviewer for this observation. Indeed, the blot shown in Supplementary Fig. 5b has now been complemented with a loading control (actin levels). Moreover, we have compared the levels of vimentin attained by transfection of RFP//vimentin with those present in several cell types in order to show that we are not working under conditions of “overexpression” (Supplementary Fig. 5a). In fact, the levels of vimentin expressed upon transfection under our experimental conditions are lower than those present, for instance, in primary fibroblasts.

This is very important. For example, the head domain of vimentin which is intact in the 1-411 construct contains many phosphorylation sites (in human over 40), and many of these sites are phosphorylated by major kinases involved in signal transduction (e.g., PKA, PKC, ROK). Therefore over expression of these domains could act as a competitive inhibitor of numerous signal transduction pathways.

We thank the reviewer for this consideration. As described above, the levels of the constructs attained by transfection in our experiments are kept at or below the levels of endogenous

vimentin in several cell types, including human fibroblasts. Nevertheless, as other effects of expressing exogenous constructs can never be completely excluded, we have included a sentence in the revised manuscript (section 7 of Results, page 13), to account for these possibilities.

In addition, although a complete deletion of the tail domain of vimentin may permit assembly of filaments, there is evidence that small deletions or point mutations in tail domains, such as the RDG sequence, actually cause defects in assembly of intermediate filaments. So justification for the use of the 1-423, 1-448, 1-459 must be rigorously tested using *in vitro* assembly assays to determine their impact on the assembly of vimentin filaments.

The question of the impact of small deletions or point mutations of vimentin tail domain on filament assembly is indeed an interesting point that has been previously addressed in the literature. However, there is not always a correlation between the assembly properties of vimentin mutants *in vitro* and the characteristics and behaviour of the filament network formed in cells. Indeed, as the reviewer points out, deletion of the tail domain permits assembly of nearly normal filaments *in vitro*; however, our results show that this mutant is not competent to form a normal network in cells on its own. In contrast, other mutations in this region, as for the G451P,Q452R mutant, have been associated with nearly normal filaments *in vitro* as well as in a substantial proportion of cells (McCormick et al., 1993, *J Cell Biol*, 122, 395). Interestingly, in Kouklis et al., 1993, *J Cell Sci*, 106, 919, internal deletion of the peptide 439-458, containing the RDG sequence, or even a “single exchange” G452V “strongly interfered with the normal assembly of IF under *in vitro* conditions”.

Therefore, following the comments of the reviewer, we found it extremely interesting to explore the cortical association of aGFP-vimentin G452V construct, based on the mutant shown by Kouklis et al., to undergo abnormal assembly *in vitro*. Remarkably, we have observed that this construct does not form squiggles or short filaments in cells but only dots, typical of defective assembly. Strikingly, this assembly-defective mutant efficiently associates with the cell cortex in mitosis (Fig. 8c). Therefore, this observation further supports the lack of correlation between assembly competence *in vitro* and cortical association in mitotic cells. In our view, this lack of correlation implies that knowledge on the competence of the truncated constructs to assemble *in vitro* would probably not add essential information about their behaviour in cells during mitosis.

We would like to stress that the aim of our paper is to monitor the behaviour of the vimentin network particularly in mitosis, a process that can only be observed in cells. Our interest is to ascertain whether the various constructs are able to translocate to the cell cortex in mitosis, independently from their assembly properties *in vitro*, and even in cells. Indeed, the purpose of the GFP-vimentin fusion proteins (Fig. 8) is to explore the cortical association of constructs that do not form full filaments in cells. In fact, besides the truncated forms, we have deliberately used vimentin constructs known to be defective in assembly (for instance, GFP-vimentin C328S, and now GFP-vimentin G452V), in order to explore the importance of assembly for this redistribution. The observation that these assembly-defective constructs efficiently redistribute to the cell periphery in mitosis, strongly supports one of the conclusions of the work, that is: full filament formation is not necessary for vimentin association with the cell cortex in mitosis.

Overall the conclusions are drawn from images of cells expressing tagged truncated vimentins and in some cases WT GFP/RFP tagged vimentin. However, many of these are of poor quality. One of numerous examples is Figure 4D.

As we have clarified above, for most of the experiments, the constructs used are not tagged, but expressed along the fluorescent proteins from bicistronic plasmids. Moreover, many experiments are performed in cells expressing endogenous vimentin. Nevertheless, we apologize for the poor resolution of some of the figures. In the revised version of the manuscript we have attempted to improve their resolution. Previous Figure 4D in particular, as well as images from videos in Fig. 2, correspond to time-lapse experiments performed with a fluorescence microscope (not confocal). In order to improve the visualization of these effects, we have used grayscale instead of color. We believe that the organization of filaments can be more clearly appreciated in these images.

The authors also fail to acknowledge that it is well known that drugs such as nocodazole alter the distribution of many types of IF including vimentin. This causes the aggregation of vimentin into large bundles and coils--so what are the side effects of long term treatments used to increase mitotic cells?

We agree with their viewer that the side effects of drugs such as nocodazole need to be taken into account. Indeed, we have used the lowest possible concentration of this compound to disrupt microtubules. In this work we have used two types of treatment: acute treatment (5 μ M for 30 min in serum-free medium), used in Fig. 3b, which completely disrupts the mitotic spindle, and partially alters microtubules in interphase cells (Fig. 3b); and "mild" long-term treatment, which is performed with 0.4 μ M nocodazole in serum-containing medium. In this latter case, although we cannot exclude all potential side effects, we have confirmed that the distribution of vimentin and actin in interphase cells is not altered, as stated in the Methods section and shown in Supplementary Fig. 1b. In addition, this treatment does not alter the actin cortex in mitosis nor the peripheral localization of vimentin.

Actually there is no convincing evidence presented that the cells depicted in Fig 5 are mitotically arrested.

We apologize for not foreseeing that this evidence could be required for former Fig. 5. Indeed, nuclei were not stained with DAPI in Fig. 5 in order not to interfere with STED. Nevertheless, mitotic cells were unequivocally identified from their typical appearance in bright field or DIC images, as described in section 5 of the Results, as well as in the experimental section of the revised manuscript. Thus, round cells without defined nuclear membrane and the rough appearance of condensed chromosomes were monitored. In the current version of Fig. 5, we now present a DIC image of an atypical mitotic cell stained with DAPI to illustrate their typical appearance characterized by the absence of defined nuclei and the presence of a central star-like pattern corresponding to condensed chromosomes. In addition, DIC images for some of the cells used for STED are provided as examples of the cells that were monitored.

The authors employ the use of many drugs such as nocodazole, latrunculin A, C3 toxin, jaspalkinolide, blebbistatin, etc. The impact of these drugs is difficult to evaluate as there appear to be no statistics; just line scans displaying fluorescence intensity. How many cells were assayed? How was the data analyzed statistically? These issues must be addressed.

We thank the reviewer for these observations. The effects of these compounds have now been statistically analyzed and the results are shown in the corresponding figures (Fig. 3g, Fig. 3h, etc.). Moreover, all the data are now provided in the Source Data file.

REVIEWERS' COMMENTS:

Reviewer #1 (Remarks to the Author):

The authors have done a good job with the revision, and with the new data, the significance of this work is brought to a higher level. I thus recommend publication. In the new Fig. 5, it would be useful to directly write out “vimentin STED” in the corresponding grayscale images. Right now this is unclear and a bit confusing.

Reviewer #2 (Remarks to the Author):

The authors have done a tremendous amount of work to address comments raised in previous reviews. In my view, they have addressed the main concerns and should not have to undergo another extensive round of revisions.

I would, however, suggest that the authors do more to emphasise the important point made clear by their new data that defects in Vimentin localisation, e.g. following expression of 1-411 or 1-423 vimentin, are associated with an accumulation of F-actin in the cytoplasm and/or around the spindle in mitosis. Note, the authors should not assume that this is equivalent to the actin cloud reported previously. It looks very different to me.

These data suggest that the localisation of Vimentin is dependent on the actin cytoskeleton and that Vimentin also affects actin organisation. While this is stated in the text, it is not part of the final Figure. In my view it now stands as one of the main findings, which they should consider mentioning in the abstract.

As the authors state in their discussion, it is also very interesting and surprising to note that Vimentin does not have to form a network to be associated with the cortex.

Finally, I would suggest the authors get go through the text carefully to correct the minor linguistic errors.

In summary, this is an interesting paper that will surprise many researchers working on mitosis and on intermediate filaments, and a wider audience of Nature communications readers.

Vimentin filaments interact with the actin cortex in mitosis allowing normal cell division

Sofia Duarte, Álvaro Viedma-Poyatos, Elena Navarro-Carrasco, Alma E. Martínez, María A. Pajares, Dolores Pérez-Sala*

RESPONSES TO REVIEWERS' COMMENTS on NCOMMS-18-28447A:

We sincerely appreciate the comments of the reviewers, which have greatly contributed to strengthen and clarify the observations in the paper.

REVIEWERS' COMMENTS:

Reviewer #1 (Remarks to the Author):

The authors have done a good job with the revision, and with the new data, the significance of this work is brought to a higher level. I thus recommend publication. In the new Fig. 5, it would be useful to directly write out “vimentin STED” in the corresponding grayscale images. Right now this is unclear and a bit confusing.

We thank the Reviewer for the positive comments. We have revised Figure 5 and its legend. We apologized for the previous lack of clarity. In the new version we have clarified that all images are STED except those in Fig. 5a and the ones that show DIC overlay, which are confocal images. We have also clarified that several versions of the same STED images are presented showing enlarged regions and/or colocalization masks. We hope that this is now clear.

Reviewer #2 (Remarks to the Author):

The authors have done a tremendous amount of work to address comments raised in previous reviews. In my view, they have addressed the main concerns and should not have to undergo another extensive round of revisions.

I would, however, suggest that the authors do more to emphasise the important point made clear by their new data that defects in Vimentin localisation, e.g. following expression of 1-411 or 1-423 vimentin, are associated with an accumulation of F-actin in the cytoplasm and/or around the spindle in mitosis. Note, the authors should not assume that this is equivalent to the actin cloud reported previously. It looks very different to me.

We sincerely thank the reviewer for the constructive criticisms and suggestions to improve the paper and strengthen its conclusions. In the revised version of the manuscript we have emphasized the aspects indicated by the reviewer. Moreover, we have included a sentence stating that the nature of the f-actin accumulation observed in the cytoplasm needs further study.

These data suggest that the localisation of Vimentin is dependent on the actin cytoskeleton and that Vimentin also affects actin organisation. While this is stated in the text, it is not part of the final Figure. In my view it now stands as one of the main findings, which they should consider mentioning in the abstract.

The final figure has been revised to emphasize the concept of vimentin-actin interplay and, in particular, the impact of vimentin on the organization of actin at the cell cortex in mitosis. Moreover, the abstract has been edited to reflect this finding.

As the authors state in their discussion, it is also very interesting and surprising to note that Vimentin does not have to form a network to be associated with the cortex.

This observation has also been stated in the abstract.

Finally, I would suggest the authors get go through the text carefully to correct the minor linguistic errors.

The text has been revised.

In summary, this is an interesting paper that will surprise many researchers working on mitosis and on intermediate filaments, and a wider audience of Nature communications readers.

EDITOR'S COMMENT:

Furthermore, we ask that you further delineate/acknowledge potential caveats in the nocodazole treatment (as mentioned from Reviewer #3 at the previous round).

Following the advice of the Editor we have introduced several considerations in the manuscript to acknowledge potential caveats in the nocodazole treatment:

In the Results section, line 343, we have introduced the following sentence:

“To increase the proportion of mitotic cells, we employed a mild nocodazole treatment. As nocodazole could affect vimentin distribution, we chose the minimum concentration to attain conditions under which actin or vimentin were not appreciably altered (see Methods and Supplementary Figure 1b).”

In Methods, line 482, the following sentence appears:

“For acute microtubule disruption, cells were treated with 5 μ M nocodazole for 30 min, whereas for mitotic arrest, cells were cultured in the presence of 0.4 μ M nocodazole for 20 h in complete medium. This treatment was employed, when indicated, to increase the proportion of mitotic cells in conditions under which neither actin, nor vimentin organization were detectably altered with respect to untreated cells and typical actin structures were preserved (Supplementary Figure 1b).”

In the legend of Fig. 9, line 1080, we have stated:

Disruption of myosin function also disrupts the cell cortex, whereas depolymerization of microtubules apparently has no effect.